

# A Multi-strategy-mode-waterlogging-prediction Framework for Urban Flood Depth

Zongjia Zhang[1,2], Jun Liang[2], Yujue Zhou[3], Zhejun Huang[2], Jie Jiang[3], Junguo Liu[4], Lili Yang[2]

[1]School of Environment, Harbin Institute of Technology, Harbin, 150001, China
[2]Department of Statistics and Data Science, Southern University of Science and Technology, Shenzhen, 518055, China
[3]Department of Computer Science and Engineering, Southern University of Science and Technology, Shenzhen, 518055, China
[4]School of Environmental Science and Engineering, Southern University of Science and Technology, Shenzhen, 518055 China

*Correspondence to*: Lili Yang (yangll@sustech.edu.cn)

**Abstract.** Flood is one of the most disruptive natural hazards, leading to massive loss of lives and considerable damage to properties. Coastal cities in Asia face floods almost every year due to monsoons influences. Early notification of flood incidents benefits the authorities and public to devise both short and long terms preventive measures, prepare evacuation and rescue missions, and relieve the flood victims. Based on time series prediction and machine learning regression algorithm, an innovative multi-strategy-mode-waterlogging-prediction framework for predicting waterlogging depth is proposed in this paper. The framework combines the historical rainfall and waterlogging depth to predict the near future waterlogging in time under future weather conditions. An expanding rainfall model was proposed to consider the positive correlation of future rainfall on the waterlogging. By selecting a suitable prediction strategy, adjusting the optimal model parameters, and then comparing the different algorithms, the optimal configuration of prediction is selected. In the actual value testing, the selected model has high computational efficiency, and the accuracy of predicting the waterlogging depth after 30 minutes can reach 86.1%, which is superior to many data-driven prediction models for waterlogging depth. The framework is helpful to timely predict the depth of target point with a high level of accuracy. It's of great significance to timely release early warning information to avoid casualties and property losses.

## 1 Introduction

With the deepening of various activities on a global scale, extreme weather and climate events occur frequently and cause a series of disasters. According to the Fifth Assessment Report of the Intergovernmental Panel on Climate Change (IPCC), global extreme weather and climate events have increased and intensified over the past 50 years, and such events will occur more frequently in the future (Jefferson, 2015). According to the Global Risk Report 2021 released by the World Economic Forum (WEF), extreme weather and climate events in 2017-2020 ranked first for four consecutive years in terms of the probability of occurrence of the top 10 global risks. Flood disaster is one of the most destructive natural disasters, usually caused by extreme weather and climate events, mainly including river basin flood, mountain flood, storm surge, urban waterlogging, and other





disaster types. Flood can cause great damage to human life, infrastructure, agriculture, and social system (Hu et al., 2021). Urban waterlogging causes casualties and huge property losses every year. From July 21 to 22, 2012, Beijing was hit by the heaviest rainstorm and waterlogging in 61 years, resulting in 79 deaths and property losses of 11.64 billion yuan. In recent years, with the development of urbanization, urban scale is expanding, population density and land utilization rate are increasing year by year, urban waterlogging caused by extreme rainstorms has become one of the serious threats to urban security. Especially with the acceleration of urbanization, a large number of low-lying areas have been included in the urban development planning. These areas are prone to waterlogging accumulation, lack of drainage capacity further aggravates the risk of waterlogging in these areas.

In the aspect of disaster prediction, most of the previous research focused on torrential and watershed floods, but the research and application of waterlogging disaster prediction were few. Scenario simulation is the main research method, which simulates the flood routing process. By constructing a hydrological dynamic model or other models to obtain the inundation area, waterlogging depth, and duration.

In the face of severe waterlogging disasters, the government needs to build a complete, reliable, and accurate waterlogging prediction and risk identification system, and develop a sustainable waterlogging risk management system that is adapted to the prediction system, focusing on early warning, prevention, rapid decision-making, and emergency rescue. Robust and accurate prediction contributes highly to water resource management strategies, policy suggestions and analysis, and further evacuation modeling (Xie et al., 2017). Thus, the importance of advanced prediction systems for flood and waterlogging is strongly emphasized to alleviate damage. However, the prediction of flood lead time is fundamentally complex due to the dynamic nature of climate conditions (Mosavi et al., 2018). The realization of timely and accurate waterlogging monitoring is of great significance for disaster assessment and prevention, timely release of early warning information to the public, targeted blockade of affected roads, and reduction of casualties and property losses caused by waterlogging (W. Wang et al., 2018).

## 2 Literature Review

### 2.1 Physically based models

Two of the most well-known Hydrodynamic Models and most used models are SWMM (Rossman 2010) and MOUSE (DHI 2016a). Conventional modeling approaches (1D and 1D-1D) can simulate quite accurately the drainage network. However, in cases of major rainfall events, these types of models are not able to simulate inundation depth in built-up areas and to visualize flood extent. Kourtis I. M. presents and assesses two different modeling approaches for the assessment of urban flooding in a small urban catchment located in the center of Athens, Greece (Kourtis et al., 2017). Yu et al. applied a 2D raster-based diffusion-wave model to determine patterns of fluvial flood inundation in urban areas using high-resolution topographic data and explores the effects of spatial resolution upon estimated inundation extent and flow routing process. The disadvantage of the 2D model is that it is difficult for the raster data model to predict the submerged area changing with time, and the performance of flow process is relatively simplified due to poor description of momentum transfer on a flood plain (D. Yu &



Lane, 2006a). Of course, its advantages are also obvious. Compared with the finite element method, finite difference method, and finite volume method, the 2D model is easy to write, with high computational efficiency and simplified calibration (D. Yu & Lane, 2006b). Abedin used SCS-CN method to estimate surface runoff, superimposed *Flow direction Grid* and *Weight Grid* to obtain *Flow length Grid*, and then obtained *Travel Time Grid* (Abedin & Stephen, 2019). Zhang et al.   presented a model using a new three-dimensional (3D) flooding model, which is an unstructured mesh, finite element model that solves the Navier-Stokes equations and developed based on Fluidity (T. Zhang et al., 2015).

Physical models showed great capabilities for predicting a diverse range of flooding scenarios, they often require various types of hydro-geomorphological monitoring datasets, requiring intensive computation. Kim et al. claims that the development of physically-based models often requires in-depth knowledge and expertise (B. Kim et al., 2015). Furthermore, although relatively fine simulation results can be obtained, comprehensive and large-scale calculations are commonly required. Therefore, it is difficult to be applied to large-scale urban flood risk identification. Y. Wang et al. used the CADDIES Flood model to identify flooded areas and constructed a flood Resilience model based on cellular automata. However, the calculation time is long and all grids need to be retraversed every time, which is not suitable for waterlogging prediction in large areas (Y. Wang et al., 2019).

## 2.2 Statistical methods

Statistical methods with advanced algorithms have been applied to identify vulnerable areas of floods. Chau et al. used K-nearest neighbors techniques to locate inundated areas in 2005 (C. Wu & Li, 2005). Hong et al. used statistical methods to draw flood-sensitive maps in 2016 D'Addabbo et al. used a statistical model based on static Bayesian networks to detect floods in 2016 (H. Hong et al., 2016). Jalayer used Bayesian parameter estimation to estimate the Topographic Wetness Index (TWI) threshold based on the inundation curve calculated by the spatial window to identify waterlogging prone areas (Jalayer et al., 2014). Falguni takes into account 9 types of factors and predicts flood through weighted superposition processing by ArcGIS (Mukherjee & Singh, 2019). The use of statistical principles to predict waterlogging requires relatively complete historical data. The advantages of this method are simple and feasible, and relatively complete factors are considered. However, it is mainly aimed at risk assessment and sensitivity analysis, so it is not effective in accurately predicting the depth of waterlogging, and usually only semi-quantitative prediction is available.

## 2.3 Data-driven models

Data-driven models can numerically formulate the flood nonlinearity, solely based on historical data without requiring knowledge about the underlying physical processes. Data-driven prediction models using machine learning are promising tools as they are quicker to develop with minimal inputs. Artificial intelligence is used to induce regularities and patterns, providing easier implementation with low computation cost, as well as fast training, validation, testing, and evaluation, with high performance compared to physical models, and relatively less complexity. In order to simulate complex mathematical



expressions of flood physical processes, machine learning methods have contributed greatly to the development of prediction
systems over the past 20 years, providing better performance and cost-effective solutions.

Machine learning has increasingly been applied in flood forecasting systems. Therein, various Machine learning strategies and
algorithms were explored. They included Decision Tree (DT) (Nayak et al., 2005), Random Forrest (RF) (Raghavendra. N &
Deka, 2014), Naïve Bayes, Artificial Neural Network (ANN) (Bellos & Tsakiris, 2016), Support Vector Machine (SVM) and
Regression (SVR) (Fotovatikhah et al., 2018) and Fuzzy Logic (Faizollahzadeh Ardabili et al., 2018). These characteristics of
the algorithms need to be clarified with respect to the type and amount of available training data, and the type of prediction
task, e.g., water level and streamflow. Amir Mosavi demonstrated the state of the art of ML models in flood prediction and
gave insight into the most suitable models, and ML models were benchmarked through a qualitative analysis of robustness,
accuracy, performance, and speed (Mosavi et al., 2018). J. Jia et al. classified urban catchment areas based on unmanned aerial
vehicle images and machine learning algorithms (Jia et al., 2022). Supattra Puttinaovarat proposed a novel flood forecasting
system based on fusing meteorological, hydrological, geospatial, and crowdsource big data in an adaptive machine learning
framework (Puttinaovarat & Horkaew, 2020). Tehrany, M.S introduced RF as an effective alternative to SVM, which often
delivers higher performance in flood prediction modeling (Tehrany et al., 2014). Bui et al. compared the performances of ANN,
SVM, and RF in general applications to floods, whereby RF delivered the best performance (Tien Bui et al., 2015). Ouyang et
al. (Ouyang et al., 2016) and Zhang et al. presented a review of the applications of ensemble ML methods used for floods.
EPSs were demonstrated to have the capability for improving model accuracy in flood modeling (J. Zhang et al., 2018).
Discrete wavelet transform (DWT) is widely applied in, e.g., rainfall-runoff (Ravansalar et al., 2017), daily streamflow
(Guimarães Santos & Silva, 2014), and reservoir inflow. The accuracy of prediction is improved through DWT, which
decomposes the original data into bands, leading to an improvement of flood prediction lead times. Neural networks have been
widely used for flood prediction in the past two decades. Kim and Barros (G. Kim & Barros, 2001) modified an ANN model
to improve flood forecasting short-term lead time. The ANN was reported to be considerably more accurate than the statistical
models. Kim, S developed an ANN forecast model for hourly lead time consisting of meteorological and hydrodynamic
parameters of three typhoons (S. Kim et al., 2016). Danso-Amoako provided a rapid system for predicting floods with an ANN,
an $R^2$ value of 0.70 for the ANN model proved that the tool was suitable for predicting flood variables with a high generalization
ability (Danso-Amoako et al., 2012). Kourgialas created a modeling system for the prediction of extreme flow based on ANNs
3 h, 12 h, and 19 h ahead of the flood, which was more effective than conventional hydrological models (Kourgialas et al.,
2015). In recent years, other improved neural networks have been gradually applied to flood prediction. Ghose, D.K. predicted
the daily runoff using a BPNN prediction model. The data of daily water-level of two years from 2013–2015 were used. The
accurate BPNN model was reported with an efficiency of 96.4% and an $R^2$ of 0.94 for flood prediction (Ghose, 2018). Chang,
F.-J. modeled multi-step urban flood forecasts using BPNN and a nonlinear autoregressive network with exogenous input
(NARX) for hourly forecasts. The results demonstrated that NARX worked better in short-term lead-time prediction compared
to BPNN. The NARX network produced an average $R^2$ value of 0.7, showing that it is effective in urban flood prediction (F.-
J. Chang et al., 2014). Bruen, M. modeled real-time rainfall-runoff forecasting for different lead times using FFNN, ARMA,



and functional networks. The models were able to predict floods with short lead times (Bruen & Yang, 2005). Lohani et al. created a rainfall-runoff modeling for water level with Adaptive Neuro-Fuzzy Inference System (ANFIS) (Lohani et al., 2014). Some studies define waterlogging prediction as a classification problem. Q Ke. defined waterlogging prediction as a binary classification problem, divided record into flood and non-flood events, and used 14 models for comparison and verification (Ke et al., 2020). Some studies used regression to predict the change of waterlogging water level in the future. Wu, Z. constructed a regression model with deep learning algorithm, named Gradient Boosting Decision Tree to predict the depth of urban flooded areas (GBDT), combined the GBDT model with hydrological variables, learned the relationship between each condition factor and the occurrence of waterlogging through training, and predicted the range and depth of waterlogging (Z. Wu et al., 2020). SVMs are more suitable for nonlinear regression problems, to identify the global optimal solution in flood models (Tehrany, Pradhan, Mansor, et al., 2015). Although the high computation cost of using SVMs and their unrealistic outputs might be demanding. SVM is used to predict a quantity forward in time based on training from past data. Over the past two decades, the SVM was also extended as a regression tool, known as support vector regression (SVR) (Li et al., 2016). Gizaw and Gan (Gizaw & Gan, 2016) developed SVR and ANN models for creating RFFA to estimate regional flood quantiles and to assess climate change impact. The SVR model estimated regional flood more accurately than the ANN model. be a suitable choice for predicting future flood under the uncertainty of climate change scenarios.

**2.4 Hybrid machine learning methods**

Most of the research used a single algorithm or model to predict, selected the optimal parameters, and worked in different datasets to test the generalization ability of the model. To improve the quality of prediction, in terms of accuracy, generalization, uncertainty, longer lead time, speed, and computation costs, there is an ever-increasing trend in building hybrid machine learning methods. These hybrid machine learning methods are numerous, including more popular ones, such as ANFIS and WNN, and further novel algorithms, e.g., FFRM–ANN (Hsu et al., 2010), ANN-hydrodynamic model, SVM–FR (Tehrany, Pradhan, & Jebur, 2015) WNN–BB, RNN–SVR (W.-C. Hong, 2008), French, J. combined machine learning with computational hydrodynamics for prediction of tidal surge inundation at estuarine ports. Procedia IUTAM 2017, 25, 28–35, SOM–R-NARX (L.-C. Chang et al., 2014), wavelet-based NARX, WBANN (Tiwari & Chatterjee, 2010), HEC–HMS–ANN (Young et al., 2017).

The application of machine learning methods to predict waterlogging disasters also has many shortcomings. If the data is scarce or does not cover varieties of tasks, their learning falls short, and hence, they cannot perform well when they are put into work. The second aspect is the capability of each ML algorithm, which may vary across different types of tasks. This can also be called a "generalization problem", which indicates how well the trained system can predict cases it was not trained for, i.e., whether it can predict beyond the range of the training dataset. For example, some algorithms may perform well for short-term predictions, but not for long-term predictions. These characteristics of the algorithms need to be clarified with respect to the type and amount of available training data, and the type of prediction task.



## 3 Methodology

### 3.1 MSMWP Framework

Accumulated rainfall is one of the most direct factors affecting the formation of waterlogging. Through data correlation analysis, we conclude that there is a certain functional relationship between rainfall and waterlogging depth, which is related to soil permeability, impervious area, air humidity, and drainage system capacity in this area.

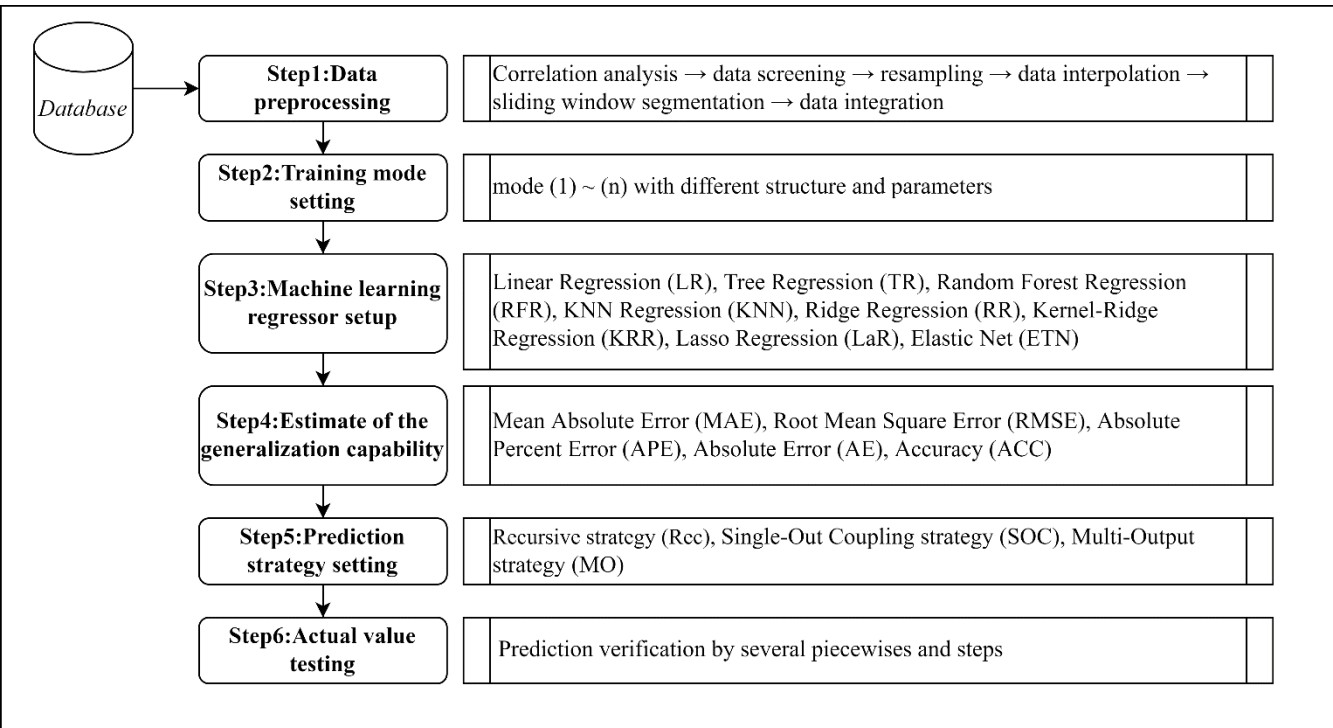

**Figure 1: The Multi-strategy-mode waterlogging prediction framework (MSMWP Framework).**

Due to the advantages of a black-box model in data-driven methods, machine learning methods can summarize these factors into an overall mechanism. Therefore, making full use of the characteristics of accumulated rainfall data will help improve the accuracy of waterlogging prediction.

To improve the accuracy of waterlogging depth prediction, this paper proposes a prediction framework (As shown in **Figure 1**) for urban waterlogging depth called MSMWP (Multi-strategy-mode waterlogging prediction) based on a variety of machine learning strategies, modes, and different algorithms for time series data. In this framework, the process of waterlogging prediction is shown as follows.



### 3.2 Working process

**Step1. Data preprocessing**

Statistical analysis, box-plot tests and correlation analysis were used to deal with missing values and outliers. Redundant data were eliminated according to the configuration conditions of the model, and an interpolation method was selected to impute the missing data after unifying the data sampling rate.

**Step2. Training mode setting**

In this paper, the accumulated rainfall data set ($R$) and the historical waterlogging depth data set ($D$) are used to predict the waterlogging depth in the future. By adjusting the data combination method $\Phi$, a new data set $X$ can be constructed by Eq. (1):

$$X = \Phi(R, D) \tag{1}$$

For each input data $x_i$, it is the vector combining $r_i (r \in R)$ and $d_i (d \in D)$ in the set sharding mode, vector $r_i$ can be represented by $[r_{i+1} \ r_{i+2} \ \cdots \ r_{i+m-1} \ r_{i+m}]$ and vector $d_i$ can be represented by $[d_{i+1} \ d_{i+2} \ \cdots \ d_{i+n-1} \ d_{i+n}]$. The sharding mode is realized by adjusting the sliding window size $m$ and $n$. The combined input vector $x_i$ can be represented as $[r_{i+1} \ r_{i+2} \ \cdots \ r_{i+m-1} \ r_{i+m} \ d_{i+1} \ d_{i+2} \ \cdots \ d_{i+n-1} \ d_{i+n}]$. Through the continuous iteration of $i$ the sliding window can loop through all the training data and combine it into the input data set $X$, (Eq. 2 to 9) which is a high-dimensional matrix.

$$R = [r_1 \ r_2 \ r_3 \ r_4 \ \cdots \ r_{l-1} \ r_l] \tag{2}$$

$$r_i = [r_{i+1} \ r_{i+2} \ \cdots \ r_{i+m-1} \ r_{i+m}], r \in R \tag{3}$$

$$D = [d_1 \ d_2 \ d_3 \ d_4 \ \cdots \ d_{l-1} \ d_l] \tag{4}$$

$$d_i = [d_{i+1} \ d_{i+2} \ \cdots \ d_{i+n-1} \ d_{i+n}], d \in D \tag{5}$$

$$x_i = [r_i \ d_i] = [r_{i+1} \ r_{i+2} \ \cdots \ r_{i+m-1} \ r_{i+m} \ d_{i+1} \ d_{i+2} \ \cdots \ d_{i+n-1} \ d_{i+n}] \tag{6}$$

$$\begin{cases} i \in (1, l - m + 1), \ m \geq n \\ i \in (1, l - n + 1), \ m < n \end{cases} \tag{7}$$

$$X = \begin{bmatrix} r_1 & r_2 & \cdots & r_m & d_1 & d_2 & \cdots & d_n \\ r_2 & r_3 & \cdots & r_{m+1} & d_2 & d_3 & \cdots & d_{n+1} \\ r_3 & r_4 & \cdots & r_{m+2} & d_3 & d_4 & \cdots & d_{n+2} \\ r_4 & r_5 & \cdots & r_{m+3} & d_4 & d_5 & \cdots & d_{n+3} \\ \vdots & \vdots & & \vdots & \vdots & & & \vdots \\ r_{i-1} & r_i & \cdots & r_{m+i-1} & d_{i-1} & d_i & \cdots & d_{n+i-1} \\ r_i & r_{i+1} & \cdots & r_{m+i} & d_i & d_{i+1} & \cdots & d_{n+i} \end{bmatrix} \tag{8}$$

$$y = \begin{bmatrix} y_1 \\ y_2 \\ \vdots \\ y_{i-1} \\ y_i \end{bmatrix} \tag{9}$$





Where $y$ is the label of the model, which stands for the output of the regressor, and it is a vector with the same length as $X$. There are five training modes under the MSMWP Framework.

**(1) Only multi-R input**

Through the analysis of data correlation, the maximum correlation coefficient between rainfall and waterlogging was 0.61. It can be concluded that there is an obvious positive correlation between rainfall and waterlogging depth., which proves that it is feasible to use accumulated rainfall to predict waterlogging.

**(2) Multi-R and single-D input**

In reality, waterlogging often occurs after raining for a period of time. Therefore, waterlogging has a certain delay characteristic
compared with rainfall. The fluctuation of waterlogging is a continuous physical process affected by multiple factors, so the waterlogging depth at the next moment is often the most closely related to the previous one. In multi-R and single-D mode, only one historical waterlogging data is selected as input.

**(3) Single-R and multi-D input**

This situation corresponds to multi-R and single-D, in which both rainfall and waterlogging data are taken into consideration
as input, but the proportion of rainfall input is reduced, while the proportion of waterlogging depth input is increased.

**(4) Multi-R and multi-D input**

This mode also covers more rainfall and waterlogging depth information, because it can better extract the characteristics of time series, balance the weight of the two data sets coupling, and better conform to the law of time change of rainfall-waterlogging.

**(5) Expanding-multi-R and multi-D**

This paper proposes a new training mode for waterlogging prediction. Because the prediction value is not only related to the past rainfall and the rainfall at the current time point, but the subsequent change of rainfall will largely affect waterlogging depth. This mode makes up for the lack of future rainfall information in mode (4) and can better reflect the dominant role of accumulated rainfall. In real applications, real-time rainfall forecast data will be added. Due to the lack of rainfall forecast data
for this area, sliding rainfall data is used as an approximation. There are two main reasons for this. Firstly, the extending part (15-30 minutes) only accounts for 12.5%-25% of the sliding rainfall (2 hours or longer) and has little effect on the whole. Second, rainfall forecasts, especially short-term forecasts of heavy rains, are now more than 90 percent accurate. It is important to note that the article does not consider only the multi-D input, because this mode building of input matrix $X$ contains only waterlogging depth information changes over time, not considering the size of the accumulated rainfall. In this mode, with the
extension of prediction time, the prediction ability of the model decreases rapidly, so it is not suitable for long-term warning. In the latter part of this paper, the results of this mode are discussed.

**Step3. Machine learning regressor setup**

Prediction of future data based on historical data is here defined as a regression problem. Because the data has the characteristics of time series, so we adopt the sliding window to slice the data in cycles. Traversal is performed in order of data



index to preserve the characteristics of continuous changes in the time dimension of the data. In this paper, eight types of regression algorithms are selected, which can simultaneously perform one-dimensional and multidimensional regression output. They are **Linear Regression (LR), Tree Regression (TR), Random Forest Regression (RFR), KNN Regression (KNN), Ridge Regression (RR), Kernel-Ridge Regression (KRR), Lasso Regression (LaR)** and **Elastic Net (ETN)**. The above eight methods are often used in the field of time series prediction. As a simple regression method, linear regression has

good applicability, but it is sensitive to outliers. The objective function of ridge regression adds the L2 regularization on the basis of the general linear regression, which ensures the best fitting error and makes the parameters as simple as possible, so that the model has strong generalization ability. X. Yu realized hydrological time series prediction using ridge regression algorithm based on feature space (X. Yu & Liong, 2007). Kernel Ridge regression algorithm had also been successfully applied to the prediction of monthly mean rainfall (Ali et al., 2020). Shen et al. took human action prediction by EEG signals as an

example to study multivariate time series prediction based on elastic net and high order fuzzy cognitive map (Shen et al., 2021). Lasso regression is normalized by adding L1 regularization after the loss function. Roy used Lasso regression to accurately predict stock market changes (Roy et al., 2015). Tree regression can model complex nonlinear data, Gocheva-Ilieva et al. developed a tree regression model to analyze and predict the time series of air pollution (Gocheva-Ilieva et al., 2019). H. Wu et al. used random forest regression algorithm to analyze the time series of weekly influenza-like incidence and made good

findings (H. Wu et al., 2017). KNN is not only used for spatial prediction, Martínez et al. proposed a time series prediction method using KNN regression algorithm in 2017 (Martínez et al., 2017).

**Step4. Evaluation**

In this paper, the evaluation is mainly divided into two stages: test stage and prediction verification stage. Among them, the indicators in the test stage mainly include the following three categories: $R^2$ score ($R^2$), Mean Absolute Error (**MAE**) and Root

Mean Square Error (**RMSE**). In the verification stage of actual value, a time series of a specific length is taken to carry out the evaluation from two parts. Firstly, in order to test the model's ability to predict the variation trend of waterlogging depth, time series covering water rising, platform and falling are intercepted. Secondly, by comparing the predicted value with the actual value, the Absolute Percent Error (**APE**) is used to calculate the model's ability of correct prediction, namely Accuracy (**ACC**). However, it is worth noting that the APE cannot be completely evaluated the model, so Absolute Error (**AE**) is needed to

supplement the evaluation. Because when the waterlogging depth is low, a large APE may correspond to a small AE. For example, the predicted value is 0.063m and the actual value is 0.056m. In this case, the APE is 12.5% and the ACC of the model is 87.5%. But the actual situation is that the predicted value is only 0.007m higher than the actual value. This error will not hinder the release of early warning information or affect the decision-making of emergency departments in the prediction of the change of waterlogging depth with a time interval of 5 minutes.


**Step5. Prediction strategy setting**

Different mode settings can greatly affect the training results of the model, and the training strategies are equally important. This paper compares three training strategies, which are Recursive, Single-Output Coupling and Multi-Output. The optimal prediction strategy is selected by comparing their performance on the test set of waterlogging prediction and the actual value test.

$Y$ can be divided into two parts, the historical data $Y_{HIS}$ and the prediction data $Y_{PRE}$. The time interval of each data set should be uniform. Even if the sensor sampling rate is different, it should be processed uniformly. We define this time interval as $\tau$ (Ben Taieb et al., 2012). In order to predict s time steps after the current time $t$, the total time of prediction is defined as $t$, where $t = \tau \times s$, according to the number of time steps covered by the time span of the desired output variable. Therefore, the set of desired output variables can be expressed as $Y_T$, $Y_T \in Y_{PRE}$. In a certain span of time $T$, between $Y_{HIS}$ and $Y_{PRE}$, we use a common notation $f^*$ in Eq. (11) to denote the functional dependency.

$$y_{pre} = f(y_{his}) + b_i \tag{10}$$

where $y_{his}$ is the inputs of each set, $y_{pre}$ stands for outputs, the functional relationship between them can be expressed as $f$, $b$ stands for modeling error, disturbances or noise.

$$Y_{PRE} = f^*(Y_{HIS}) + \sum b_i \tag{11}$$

**i. Recursive strategy (Rec)**

The oldest and most intuitive forecasting strategy is the Recursive (also called Iterated or Multi-Stage) strategy (Ji et al., 2005). The result of the prediction of the first step is embedded into the last element of the input vector of the next prediction, and the result of the prediction of the second step is obtained. In Eq. (12), when s=1, $y_t, y_{t-1}, \cdots, y_{t-n}$ can be used to predict $\hat{y}_{t+1}$. When s=2, the previous predicted value $\hat{y}_{t+1}$ is used to replace the first element of the input vector, and the following elements are replaced in turn. The original last element $y_{t-n}$ is removed from the input vector. The model is iterated recursively, and the mixed vector of historical data and forecast data is used as the model input. The algorithm for this method can be expressed as **Algorithm 1**. The prediction vector $\hat{\boldsymbol{y}}_{\boldsymbol{pre}}[\boldsymbol{p}]$ in the first step can be obtained based on historical data. Then the final element of the input vector $X_{pre}[p][d_s - 1]$ is replaced by $\hat{\boldsymbol{y}}_{\boldsymbol{pre}}[\boldsymbol{p} - \boldsymbol{1}]$ and the new $\hat{\boldsymbol{y}}_{\boldsymbol{pre}}$ is obtained through the input model. And so on, move the value of $X$ forward one bit and add the new predicted value.

$$\hat{y}_{t+s} = \begin{cases} \hat{f}(y_t, y_{t-1}, \cdots, y_{t-n}), & s = 1 \\ \hat{f}(\hat{y}_{t+s-1}, y_{t+s-2}, \cdots, y_{t+s-n}), & s = 2 \\ \quad\quad\quad \vdots \\ \hat{f}(\hat{y}_{t+s-1}, \cdots, \hat{y}_{t+s-n}), & s \in (2, n] \end{cases} \tag{12}$$



---

**Algorithm 1**. Recursive Strategy

**Input:** $D = \{X, Y_{HIS}\}$, Training Dataset.

**Input:** $Reg = \{Regressor^{(n)}\}$, **n** kinds of regression method.

**Input:** $M = \{mode(1\ to\ 5)\}$, Mode.

**Output:** $\hat{y} = \{\hat{y}_{pre}\}$, the prediction of the output of the selected series.

**1** Using training data, different models and regressors were selected for model training.

**2 for** $k \in (1, N)$ **do,** test model for N times, select the best output.

**3 for** $p \in \{1, s\}$ **do**.

**4** $\quad$ $d_s = len(\boldsymbol{x_i})_{x_i \in \{X\}}$, recursive index

**5** $\quad$ $APE = \left| \dfrac{\hat{y}_{pre}^{(p)} - y^{(p)}}{y^{(p)}} \right|$

**6** $\quad$ $AE = \left| \hat{y}_{pre}^{(p)} - y^{(p)} \right|$

**7** $\quad$ $MSE = \sum_m^1 (y - \hat{y})$

**8** $\quad$ $e(k) = \{APE^{(k)}, AE^{(k)}, MSE^{(k)}\}$, comprehensive error evaluation.

**9** $\quad$ $\boldsymbol{\hat{y}_{pre}}[\boldsymbol{p}] = model.predict(X_{pre}[p])$

**10** $\quad$ $X_{pre}[p][d_s - s - 1] = X_{pre}[p - 1][d_s - s]$

**11** $\quad$ ......... ............ ........

**12** $\quad$ $X_{pre}[p][d_s - 2] = X_{pre}[p - 1][d_s - 1]$

**13** $\quad$ $X_{pre}[p][d_s - 1] = \boldsymbol{\hat{y}_{pre}}[\boldsymbol{p - 1}]$

**14 end**

**15** $k = arg\ min_{k \in \{1 \cdots N\}} e(k)$.

**16** $\hat{y} = \{y_{pre}^{(k)}\}$.

**17 return** $\hat{y}$.

---

## ii. Single-Out Coupling strategy (SOC)

The Single Output Coupling strategy is similar to the Direct strategy proposed by Cheng et al. (Cheng et al., 2006) and
Hamzacebi et al (Hamzaçebi et al., 2009). Different machine learning models have been used to implement the Direct strategy
for multi-step ahead forecasting tasks, for instance neural networks (Kline, 2004), nearest neighbors (Sorjamaa et al., 2007)
and decision trees (Guimarães Santos & Silva, 2014). It consists of forecasting each horizon independently from the others.
The biggest difference from Recursive strategy is that Single-Out Coupling does not use any approximated values to compute
the forecasts, so having no accumulation of errors. In this strategy, error of the previous prediction results will not have a great
influence on the later prediction results. Each $f$ is supported by a corresponding model and trained with its own independent
data (Eq. 13). When s =1, it is the same as one-step prediction. When s>1, the model makes prediction across the time interval
of s steps. Finally, the results of Single-Out $y_{t+1}, y_{t+2}, \cdots, y_{t+s-1}, y_{t+s}$ coupling by Eq. (14) into a new forecast time
series$[y_{t+1}, y_{t+2}, \cdots, y_{t+s-1}, y_{t+s}]$.

$$y_{t+s} = f_s(y_t, y_{t-1}, \cdots, y_{t-n+1}) \tag{13}$$





$$\begin{cases} y_{t+1} = f_1(y_t, y_{t-1}, \cdots, y_{t-n+1}) + b_i \\ y_{t+2} = f_2(y_t, y_{t-1}, \cdots, y_{t-n+1}) + b_i \\ \qquad\qquad \vdots \\ \qquad\qquad \vdots \\ y_{t+s} = f_s(y_t, y_{t-1}, \cdots, y_{t-n+1}) + b_i \end{cases} \qquad (14)$$

### iii. Multi-Output strategy (MO)

The two previous strategies (Recursive and Single-Out Coupling) may be considered as Single-Output strategies, which neglects the existence of stochastic dependencies between future values and consequently affects the forecast accuracy. Multi-Output strategy requires the design of Multiple-Response modeling techniques. The output is no more a scalar quantity but a

vector of length $s$. Using only one model, output a time series of s time intervals (Eq. 15). Compared with Single-Output Coupling strategy, this strategy has simple operation and fast calculation. The disadvantage is that some regression algorithms such as Bayesian regression, GBRT and AdaBoosting, do not support multidimensional output directly.

$$[y_{t+s}, y_{t+s-1}, \cdots, y_{t+2}, y_{t+1}] = f_m(y_t, y_{t-1}, \cdots, y_{t-n+2}, y_{t-n+1}) + \vec{b_i} \qquad (15)$$

Where $f_m$ is vector-valued function. $b_i$ is a noise vector.

**Step6. Actual value testing**

In order to prevent over-fitting phenomenon or insufficient prediction of trend change, it is necessary to test the prediction performance of the framework in actual waterlogging data sets after completing training and testing. N groups of continuous time series are selected for actual value test and the application results in actual data will be discussed.

### 4 Case Study

### 4.1 Research area and objectives

As an important city in south China, Shenzhen is one of the core cities of the Guangdong-Hong Kong-Macao Greater Bay Area. As a representative of China's special economic zones, Shenzhen's urban construction and development has been attracting much attention since its establishment 40 years ago. Its economic growth rate has grown rapidly from 270 million yuan in 1980 to 2,767.024 billion yuan in 2021.

In the process of rapid urban development, Shenzhen is also facing many challenges from natural disasters and accidents, which often bring serious threats to urban public safety and people's health. Shenzhen, located in the southeast coast of China, has a subtropical monsoon climate. Influenced by the Pacific monsoon current, it receives sufficient rainfall all year round and is greatly influenced by the Pacific typhoon. The rainfall is unusually concentrated from June to September every year, and heavy rains or extremely heavy rains occur frequently. In particular, the frontal rainfall in Shenzhen area is subject to

topography, which often forms local sudden rainstorms with short duration. It usually takes 8 to 12 hours of rainfall to reach the normal 24 hours of rainfall. According to the statistics of rainfall data in Shenzhen from 1960 to 2012, there were 8.8 times

of heavy rainfall with daily rainfall of more than 50mm, 75.2% of which were heavy rain, 21.4% of which were torrential rain, and 3.4% of which were extremely torrential rain (Chen et al.,2020). In May 2014, Pingshan District, Shenzhen, was hit by a sudden rainstorm, with 261mm of rainfall in three hours, causing 150 houses to be flooded and 2,600 people affected. The

rainstorm event on April 11, 2019 saw the heaviest rainfall in April since Shenzhen began meteorological records, causing floods in several regions and 11 deaths (Liu et al., 2020).

This study focuses on the areas vulnerable to waterlogging in Shenzhen, integrates meteorological and hydrological data from Shenzhen city database, and uses machine learning methods to analyze and process historical meteorological data and waterlogging sensor water level data. In this study, a data-driven prediction model of urban rainstorm waterlogging depth is

established, which can realize the advance perception and accurate prediction of water level change of a waterlogging point.

**Figure 2: Waterlogging sensors and weather stations in Shenzhen.**



**Figure 3: Rainfall region division obtained by Tyson polygon algorithm in Shenzhen.**

## 4.2 Data preprocessing

There are two main parts of the data used in the study. The first part is the meteorological observation data of Shenzhen provided by the Shenzhen Meteorological Bureau. The data covers the time range from March 8th, 2019, to August 17th July 18th, 2020, including the meteorological observation data of rainfall, wind speed, visibility, temperature and humidity at 242 stations in the city. The second part is the waterlogging depth sensor data of 170 observation stations in the city provided by Shenzhen Municipal Water Bureau, with an accuracy of 1 cm. Changes in waterlogging depth affect parameters such as refractive index and pressure. The sensor senses the changes and converts physical signals into electrical signals, which are transmitted to the database through optical fibers. The biggest time range is from January 1st, 2019, to July 18th, 2020.

Observation points of meteorological data and waterlogging depth data cover all 10 districts of Shenzhen, and their spatial distribution is shown in the **Figure 2**. Tyson polygon algorithm is used to divide regions according to the geographical location of meteorological stations, and the rainfall coverage of each station is obtained. The polygon surface of each region indicates





that meteorological data of this station is used in this region (**Figure 3**) (Luo et al., 2020). Through this classification form, it is determined that 170 observation stations of waterlogging depth have unique corresponding meteorological input, which unified the model input and output relationship in spatial dimension.

Because waterlogging sensors are set up in batches, the total running time and data storage capacity of each sensor are different. Among 170 waterlogging sensors, we selected the sensor_123 with a long operation time and a large amount of data as the research object. Through data analysis and consistency test of meteorological data, it is concluded that rainfall is the most critical factor affecting waterlogging depth. Rainfall data include sliding rainfall with different window lengths, *R10M R30M, R01H R02H, R03H, R06H, R12H, R24H* and *R72H* for 10 minutes, 30 minutes, 1 hour, 2 hours, 3 hours, 6 hours, 12 hours, 24

hours and 72 hours of sliding rainfall values. D means the waterlogging depth. Through the data correlation analysis between sliding rainfall and waterlogging depth data of each station (**Figure 4**), it is concluded that *R02H* has the largest correlation degree of **0.61** with waterlogging depth. In the **Figure 4**, the darker the color, the lower the correlation, and the lighter the color, the higher the correlation.

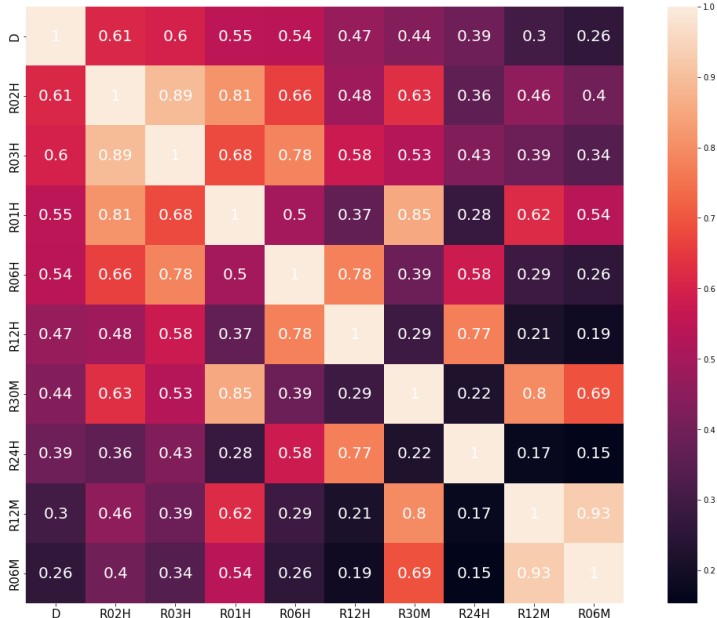

**Figure 4: Data correlation analysis of rainfall data at weather station_ G3795 and waterlogging depth data at sensor_123, the maximum correlation coefficient with D is 0.61 (D and R02H) and the minimum is 0.26 (D and R06M).**

**4.3 Data integration**

Due to the special working mechanism of the waterlogging depth sensor, the data sampling rate is not uniform. In the period when there is no water accumulation (the waterlogging depth is 0), it is collected at irregular intervals of several hours or even

several days. In the period when the water level changes dramatically, the sensor can collect data once a minute at the fastest, which brings some difficulties to our research. Considering that the interval of rainfall data is 5 minutes, in order to balance the model accuracy and training efficiency, the data of waterlogging depth is resampled first, which is consistent with the




rainfall data on the time scale. Then data interpolation is performed on the newly added blank interval of resampling, which

does not destroy the original characteristic attributes of the data. The fluctuation process of waterlogging is a smooth and

continuous process, reflected in the graph curve is smooth. Five commonly used interpolation methods are tested in this paper,

they Cubic, Quadratic, Linear, Zero and Nearest. The optimal interpolation method can be determined by comparing the MAPE

of the interpolation data $Y_{insert}$ and the actual data $Y_{true}$ and observing the fitting of the interpolation curve and the actual

curve. Finally, the Linear interpolation method is applied, and the interpolation form is internal interpolation. (Cubic and

Quadratic may have negative values, while Zero and Nearest have obvious ladder characteristics, which are not consistent with

the continuous change characteristics of the waterlogging). After analysis, a total of 527 non-zero interpolation data, accounting

for 0.46% of the total data set (143,424). Interpolation data are mainly concentrated at the beginning and end of the water, and

the values are generally low. This part of data preprocessing unified the model input and output relationship in the time

dimension, and the time range of the final rainfall and waterlogging depth data was unified from 00:00:00 on March 8th, 2019,

to 23:55:00 on July 18th, 2020, with an interval of 5 minutes.

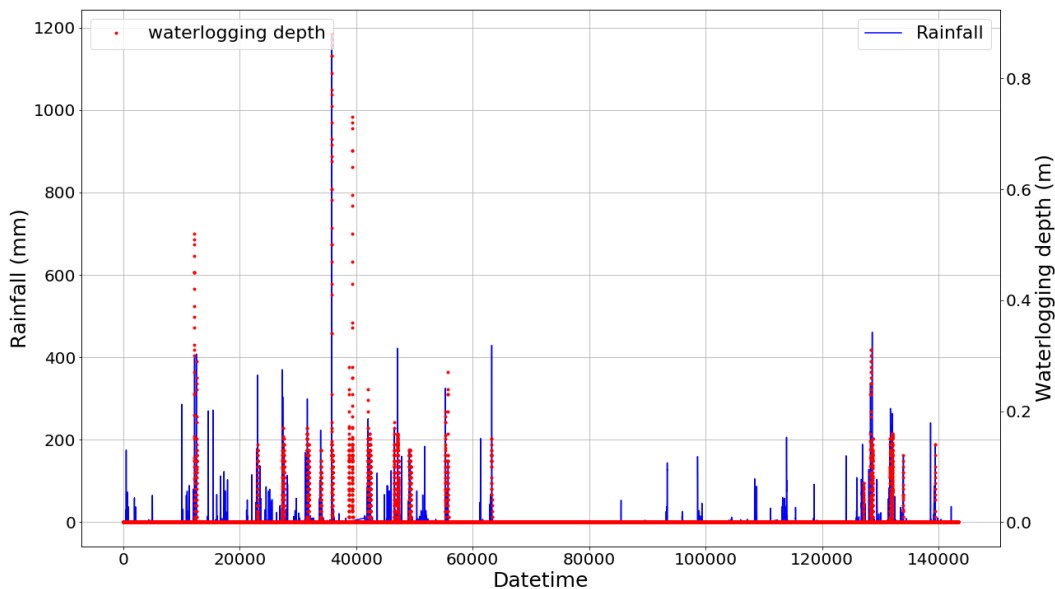


**Figure 5: Rainfall and waterlogging from 00:00:00 on March 8th, 2019, to 23:55:00 on July 18th, 2020 (with an interval of 5mins).**

After determining the location correlation between rainfall stations and waterlogging observation stations and unifying the

time scale, the two data were integrated into one data. From **Figure 5**, it can be seen that in most of the time interval, when the

rainfall is 0, the frequency of waterlogging accumulation data is less. Through data visualization, it can be known that there is

no rainfall in most time interval, and the frequency of waterlogging accumulation data is less, which is consistent with the

reality. Although the proportion of impervious water surface in urban construction area increases year by year, the frequency

of waterlogging accumulation caused by surface runoff has decreased due to the continuous improvement of drainage system

construction and the application of sponge city engineering. However, in the event of strong typhoon or heavy rain, the drainage





volume still cannot meet the needs of urban drainage. This would overload the drainage system and allow large amounts of

urban surface runoff to accumulate in low-lying areas. In this study, considering the factors of surface infiltration, vegetation

leaf canopy interception effect and evaporation, surface runoff cannot be formed under extremely low rainfall and non-rainfall,

so the waterlogging depth is always 0. In order to avoid the interference of such factors on model training, the minimum rainfall

threshold is set here as 5 millimeters. By searching the entire data set, lock the start and end time stamps of each rainfall event

interval, named *R_STA* and *R_END*, Rainfall duration is denoted by *R_DUR*. In the entire data set, *R_STA* and *R_END* are

paired to represent the rainfall start and end index. A total of 251 rainfall event time series (982.34 hours in total, average

rainfall duration is 3.91 hours) were obtained by screening 143,424 data points from this site, and a new dataset *Rain_Set* with

12,309 data points (**Figure 6**) was constructed. This method eliminates the interference of a large number of sunny weather

inputs to the model, which is proved to improve the efficiency and accuracy of the model calculation.

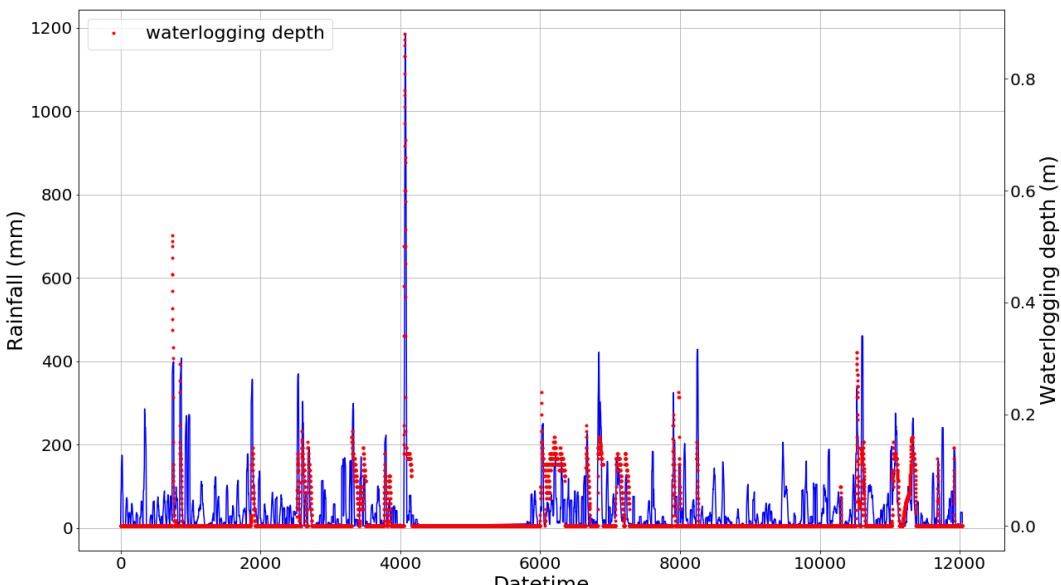

**Figure 6: Rainfall (blue solid line) and waterlogging (red dotted line) according to Rain_Set.**

**4.4 Research methods**

**4.4.1 Construction of model data sets**

Based on MSMWP framework, after data preprocessing, different training modes are constructed by changing $\phi$ to adjust

$R \ and \ D$ combination mode. Each rainfall time series is processed by cyclically cutting in the form of sliding windows, we

can get a matrix $R$ as Eq. (16). In the five modes, input vectors of different dimensions can be constructed by adjusting the

values of **m** and **n**, and multiple models would be trained. The goal of the model is to accurately predict the waterlogging depth

at a certain time in the future. Assume that the current time is $t$, and the predicted target value of the waterlogging depth in

the future is $y_{t+1}, y_{t+2}, y_{t+3}, \cdots, y_{t+s-1}, y_{t+s}$. When **m=6** and **n=1**, it means that the model selects rainfall in 30 minutes and





waterlogging in 5 minutes before time $t$ as input for training. When **m=12** and **n=3**, it means that the model selects the rainfall

in 1h and waterlogging in 15minutes before time t as input for training. Under different combination conditions of **m** and **n**,

the five combination modes are selected as shown in **Table 1**:

Table 1: Mode (1) to (5) with different parameter m and n.

| (1) Only multi-R input | (2) Multi-R and single-D input | (3) Single-R and multi-D input | (4) Multi-R and multi-D input | (5) Expanding-multi-R and multi-D |
|---|---|---|---|---|
| m=6 | m=6, n=1 | m=1, n=6 | m=6, n=3 | m=6(3:3), n=3 |
| m=12 | m=12, n=1 | m=1, n=12 | m=12, n=3 | m=12(9:3), n=3 |
| m=18 | m=18, n=1 | m=1, n=18 | m=18, n=3 | m=18(12:6), n=3 |
| m=24 | m=24, n=1 | m=1, n=24 | m=24, n=3 | m=24(18:6), n=3 |
| —— | —— | —— | m=6, n=6 | m=6(3:3), n=6 |
| —— | —— | —— | m=12, n=6 | m=12(9:3), n=6 |
| —— | —— | —— | m=18, n=6 | m=18(12:6), n=6 |
| —— | —— | —— | m=24, n=6 | m=24(18:6), n=6 |

$$Rainfall\ set: R = \begin{bmatrix} [\ 5, 18, 41, 63, \cdots, 97, 99, 130, 147] \\ [18, 41, 63, 74 \cdots 99, 130, 147, 151] \\ [41, 63, 74, 83 \cdots 130, 147, 151, 158] \\ [\cdots \cdots \cdots \cdots \cdots \cdots \cdots \cdots \cdots] \\ [\ 6, 4, 2, 1, \cdots \cdots \cdots \cdots, 0, 0, 0, 0] \\ [\ 4, 2, 1, 1, \cdots \cdots \cdots \cdots, 0, 0, 0, 0] \end{bmatrix} \tag{16}$$

Among them, the first three modes are relatively simple. Since rainfall is the fundamental factor affecting waterlogging, the

dimension of rainfall in the latter two modes is basically higher than that of waterlogging. In Expanding- multi-R and multi-

D, the rainfall input is split with t as the dividing line to emphasize the influence of the subsequent continuous rainfall input

on the model. For example, **m=12(9:3)** means that rainfall 45 minutes before t and 15 minutes after t are selected as input.

The strategy in step 5 influences the label selection of the model. The Recursive strategy requires direct prediction of $y_{t+1}$

and then recursion with label $y_{t+1}$.

Different modes, strategy, machine learning regressor together affected the training and prediction results of the model. Single-

Out Coupling strategy outputs 5 minutes, 10 minutes until the moment $(s \times 5)$ minutes of waterlogging, so the label is

$y_{t+1}, y_{t+2}, y_{t+3}, \cdots, y_{t+s}$. Multi-Output strategy outputs the waterlogging value vector of minutes at $(5\ \ to\ s \times 5)$ at the same

time, so its label is $[y_{t+1}, y_{t+2}, y_{t+3}, \cdots, y_{t+s}]$. In this paper, eight regression algorithms are selected, which can simultaneously

perform one-dimensional and multidimensional regression output.

**4.4.2 Training set and testing set**

The data set constructed according to the above combination method was divided into training set and test set at a ratio of 70%

and 30%. Different modes, strategies and regression algorithms were applied for training, and evaluated by **RMSE, MAE** and

**R² score**. The curve fitting of the predicted data on the test set was compared with the actual data, and the test results of each

configuration were analyzed and sorted.

$MAE = \frac{1}{m} \sum_{i=1}^{m} |\hat{y}^{(i)} - y^{(i)}| \tag{17}$



$$RMSE = \sqrt{\frac{1}{m}\sum_{i=1}^{m}(\hat{y}^{(i)} - y^{(i)})^2} \tag{18}$$

$$R^2 = 1 - \frac{\sum_i(\hat{y}^{(i)} - y^{(i)})^2}{\sum_i(\bar{y} - y^{(i)})^2} \tag{19}$$

### 4.5 Result and discussion

This section presents and discusses the testing results of the different mode and forecasting strategies. For each mode, we
report the results obtained in the eight different regression methods. Based on the results of actual value verification, different
prediction strategies are discussed.

### 4.5.1 Testing result

The time series intercepted with rainfall events were integrated into a new data set, with a total of 12,039 data points. The first
70% constituted a training set with 8,428 points of data, and the last 30% constituted a testing set with 3,611 points of data. In
the testing set, eight regression methods were used to test 5 modes, and the model structure inside each mode was changed by
adjusting the input parameters **m** and **n**. The testing results of different modes are shown in **Table 2** to **Table 6.** The numbers
in brackets represent the model evaluation ranking under different combinations of m and n parameters for a certain regression
method, and the bold characters represent the optimal results. The underlined indicates that MAE, RMSE, and $R^2$score rank in
the top 50% of all methods in the best results (bold characters) within the method. Taking mode (3)-KRR as an example, in
KRR optimal indicator, RMSE is 0.0051, ranking the second, MAE is 0.0013, ranking the third, $R^2$score is 0.9779, ranking the
third, so the underlined indicator of KRR is 3.





**Table 2: Testing result of mode (1) Only multi-R input.**

| Mode | Evaluation Indicator | Regression methods | | | | | | | |
|---|---|---|---|---|---|---|---|---|---|
| | | LR | TR | RFR | KNN | RR | KRR | LaR | ETN |
| m=6 | MAE | 0.0191(4) | 0.0191(2) | 0.0189(4) | 0.0177(4) | 0.0191(4) | 0.0204(4) | 0.0199(4) | 0.0191(4) |
| | RMSE | 0.0325(4) | 0.0434(2) | 0.0338(4) | **0.0337(1)** | 0.0325(4) | 0.0329(4) | 0.0306(4) | 0.0312(4) |
| | R$^2$score | 0.1209(4) | **-0.5649(1)** | 0.0474(4) | **0.0540(1)** | 0.1209(4) | 0.1008(4) | 0.2206(3) | 0.1905(4) |
| m=12 | MAE | 0.0181(3) | 0.0198(3) | 0.0181(3) | 0.0174(2) | 0.0181(3) | 0.0191(3) | 0.0194(3) | 0.0186(3) |
| | RMSE | 0.0310(3) | 0.0444(4) | 0.0324(3) | 0.0347(2) | 0.0310(3) | 0.0313(3) | 0.0300(2) | 0.0304(3) |
| | R$^2$score | 0.1800(3) | -0.6803(3) | 0.1078(2) | -0.0264(2) | 0.1800(3) | 0.1632(3) | 0.2339(2) | 0.2133(3) |
| m=18 | MAE | 0.0168(2) | 0.0198(3) | 0.0180(2) | 0.0174(2) | 0.0168(2) | 0.0175(2) | **0.0191(1)** | 0.0181(2) |
| | RMSE | 0.0286(2) | 0.0442(3) | 0.0323(2) | 0.0366(4) | 0.0286(2) | 0.0289(2) | **0.0293(1)** | 0.0290(2) |
| | R$^2$score | 0.2808(2) | -0.7160(4) | 0.0861(3) | -0.1730(4) | 0.2808(2) | 0.2666(2) | **0.2465(1)** | 0.2648(2) |
| m=24 | MAE | **0.0156(1)** | **0.0189(1)** | **0.0168(1)** | **0.0168(1)** | **0.0156(1)** | **0.0163(1)** | **0.0191(1)** | **0.0175(1)** |
| | RMSE | **0.0261(1)** | **0.0432(1)** | **0.0307(1)** | 0.0359(3) | **0.0261(1)** | **0.0264(1)** | 0.0301(3) | **0.0276(1)** |
| | R$^2$score | **0.3954(1)** | -0.6576(2) | **0.1654(1)** | -0.1438(3) | **0.3954(1)** | **0.3825(1)** | 0.1980(4) | **0.3269(1)** |
| Underlined indicator | | 3 | 0 | 0 | 0 | 3 | 3 | 0 | 0 |


**Table 3: Testing result of mode (2) Multi-R and single-D input.**

| Mode | Evaluation Indicator | Regression methods | | | | | | | |
|---|---|---|---|---|---|---|---|---|---|
| | | LR | TR | RFR | KNN | RR | KRR | LaR | ETN |
| m=6 n=1 | MAE | 0.0013(4) | 0.0016(4) | 0.0011(2) | **0.0167(1)** | 0.0022(2) | **0.0020(1)** | 0.0199(4) | 0.0191(4) |
| | RMSE | 0.0044(4) | 0.0079(4) | **0.0055(1)** | **0.0331(1)** | 0.0050(4) | 0.0050(4) | 0.0306(4) | 0.0312(4) |
| | R$^2$score | 0.9842(4) | 0.9477(4) | **0.9731(1)** | **0.0917(1)** | 0.9791(4) | 0.9792(4) | 0.2206(3) | 0.1905(4) |
| m=12 n=1 | MAE | 0.0012(3) | 0.0015(3) | 0.0012(4) | 0.0171(3) | **0.0021(1)** | **0.0020(1)** | 0.0194(3) | 0.0186(3) |
| | RMSE | 0.0043(3) | 0.0069(3) | **0.0057(1)** | 0.0346(2) | 0.0049(3) | 0.0049(3) | 0.0300(2) | 0.0304(3) |
| | R$^2$score | 0.9845(3) | 0.9591(2) | 0.9721(2) | -0.0167(2) | 0.9794(3) | 0.9795(3) | 0.2339(2) | 0.2133(3) |
| m=18 n=1 | MAE | 0.0011(2) | 0.0013(2) | 0.0011(2) | 0.0174(4) | 0.0022(2) | **0.0020(1)** | **0.0191(1)** | 0.0181(2) |
| | RMSE | 0.0036(2) | 0.0068(2) | 0.0067(4) | 0.0365(4) | **0.0046(1)** | **0.0046(1)** | **0.0293(1)** | 0.0290(2) |
| | R$^2$score | 0.9888(2) | 0.9589(3) | 0.9606(4) | -0.1713(4) | **0.9815(1)** | **0.9815(1)** | **0.2465(1)** | 0.2648(2) |
| m=24 n=1 | MAE | **0.0010(1)** | **0.0010(1)** | **0.0010(1)** | **0.0167(1)** | 0.0024(4) | 0.0023(4) | **0.0191(1)** | **0.0175(1)** |
| | RMSE | **0.0035(1)** | **0.0055(1)** | 0.0058(3) | 0.0359(3) | 0.0048(2) | 0.0047(2) | 0.0301(3) | **0.0276(1)** |
| | R$^2$score | **0.9894(1)** | **0.9730(1)** | 0.9706(3) | -0.1410(3) | 0.9800(2) | 0.9801(2) | 0.1980(4) | **0.3269(1)** |
| Underlined indicator | | 3 | 2 | 3 | 0 | 2 | 3 | 0 | 0 |





**Table 4: Testing result of mode (3) Single-R and multi-D input.**

| Mode | Evaluation Indicator | Regression methods | | | | | | | |
|---|---|---|---|---|---|---|---|---|---|
| | | LR | TR | RFR | KNN | RR | KRR | LaR | ETN |
| m=1 n=6 | MAE | 0.0008(4) | **0.0014(1)** | 0.0010(2) | 0.0073(4) | **0.0016(1)** | 0.0016(4) | 0.0199(4) | 0.0191(4) |
| | RMSE | 0.0039(3) | 0.0073(2) | **0.0052(1)** | 0.0233(4) | **0.0052(1)** | **0.0051(1)** | 0.0306(4) | 0.0312(4) |
| | R²score | 0.9871(3) | 0.9558(2) | **0.9797(1)** | 0.5474(4) | **0.9779(1)** | **0.9779(1)** | 0.2206(3) | 0.1905(4) |
| m=1 n=12 | MAE | 0.0007(2) | **0.0014(1)** | 0.0011(4) | 0.0073(4) | **0.0016(1)** | 0.0014(3) | 0.0194(3) | 0.0186(3) |
| | RMSE | 0.0040(4) | 0.0076(3) | 0.0062(4) | 0.0226(3) | 0.0055(2) | 0.0055(2) | 0.0300(3) | 0.0304(3) |
| | R²score | 0.9866(4) | 0.9512(3) | 0.9674(3) | 0.5669(3) | 0.9745(2) | 0.9746(2) | 0.2339(2) | 0.2133(3) |
| m=1 n=18 | MAE | 0.0007(2) | 0.0015(4) | 0.0010(2) | 0.0071(2) | 0.0017(3) | **0.0013(1)** | **0.0191(1)** | 0.0182(2) |
| | RMSE | **0.0029(1)** | 0.0079(4) | 0.0062(4) | 0.0220(2) | 0.0057(3) | 0.0056(3) | **0.0293(1)** | 0.0291(2) |
| | R²score | **0.9925(1)** | 0.9447(4) | 0.9665(4) | 0.5742(2) | 0.9718(3) | 0.9721(3) | **0.2467(1)** | 0.2558(2) |
| m=1 n=24 | MAE | **0.0006(1)** | **0.0014(1)** | **0.0009(1)** | **0.0070(1)** | 0.0018(4) | 0.0014(3) | 0.0192(2) | **0.0180(1)** |
| | RMSE | **0.0029(1)** | **0.0068(1)** | 0.0053(2) | **0.0217(1)** | 0.0058(4) | 0.0057(4) | 0.0298(2) | **0.0281(1)** |
| | R²score | **0.9925(1)** | **0.9594(1)** | 0.9750(2) | **0.5828(1)** | 0.9705(4) | 0.9710(4) | 0.2143(4) | **0.2993(1)** |
| Underlined indicator | | 3 | 1 | 3 | 0 | 2 | 3 | 0 | 0 |




**Table 5: Testing result of mode (4) Multi-R and multi-D input.**

| Mode | Evaluation Indicator | Regression methods | | | | | | | |
|------|------|------|------|------|------|------|------|------|------|
| | | LR | TR | RFR | KNN | RR | KRR | LaR | ETN |
| m=6 n=3 | MAE | 0.0010(7) | 0.0012(2) | **0.0010(1)** | **0.0166(1)** | 0.0022(7) | 0.0020(7) | 0.0199(7) | 0.0191(7) |
| | RMSE | 0.0039(5) | 0.0062(4) | **0.0049(1)** | **0.0330(1)** | 0.0058(4) | 0.0058(5) | 0.0306(7) | 0.0312(7) |
| | R²score | 0.9872(5) | 0.9684(4) | **0.9803(1)** | 0.0918(2) | 0.9723(3) | 0.9725(4) | 0.2206(5) | 0.1905(7) |
| m=12 n=3 | MAE | **0.0008(1)** | 0.0012(2) | **0.0010(1)** | 0.0171(5) | 0.0021(4) | 0.0019(5) | 0.0194(5) | 0.0186(5) |
| | RMSE | 0.0039(5) | 0.0060(3) | 0.0057(3) | 0.0346(3) | 0.0058(4) | 0.0058(5) | 0.0300(3) | 0.0304(5) |
| | R²score | 0.9870(7) | 0.9692(3) | 0.9726(3) | -0.0167(4) | 0.9713(6) | 0.9715(6) | 0.2339(3) | 0.2133(5) |
| m=18 n=3 | MAE | **0.0008(1)** | 0.0012(2) | **0.0010(1)** | 0.0174(7) | **0.0019(1)** | **0.0017(1)** | **0.0191(1)** | 0.0181(3) |
| | RMSE | **0.0029(1)** | 0.0065(5) | 0.0068(8) | 0.0365(7) | 0.0055(2) | 0.0055(2) | **0.0293(1)** | 0.0290(3) |
| | R²score | **0.9924(1)** | 0.9628(6) | 0.9593(8) | -0.1713(7) | 0.9723(3) | 0.9735(2) | **0.2465(1)** | 0.2648(3) |
| m=24 n=3 | MAE | **0.0008(1)** | **0.0010(1)** | **0.0010(1)** | 0.0167(3) | **0.0019(1)** | **0.0017(1)** | **0.0191(1)** | **0.0175(1)** |
| | RMSE | **0.0029(1)** | **0.0050(1)** | 0.0060(5) | 0.0359(5) | **0.0054(1)** | **0.0054(1)** | 0.0301(5) | **0.0276(1)** |
| | R²score | **0.9924(1)** | **0.9778(1)** | 0.9678(6) | -0.1410(5) | **0.9739(1)** | **0.9742(1)** | 0.1980(7) | **0.3269(1)** |
| m=6 n=6 | MAE | 0.0010(7) | 0.0013(5) | **0.0010(1)** | **0.0166(1)** | 0.0022(7) | 0.0020(7) | 0.0199(7) | 0.0191(7) |
| | RMSE | 0.0039(5) | 0.0071(8) | **0.0049(1)** | **0.0330(1)** | 0.0057(3) | 0.0057(3) | 0.0306(7) | 0.0312(7) |
| | R²score | 0.9871(6) | 0.9586(8) | 0.9796(2) | **0.0920(1)** | 0.9730(2) | 0.9733(3) | 0.2206(5) | 0.1905(7) |
| m=12 n=6 | MAE | **0.0008(1)** | 0.0013(5) | **0.0010(1)** | 0.0171(5) | 0.0020(3) | 0.0018(3) | 0.0194(5) | 0.0186(5) |
| | RMSE | 0.0039(5) | 0.0067(7) | 0.0060(5) | 0.0345(3) | 0.0058(4) | 0.0057(3) | 0.0300(3) | 0.0304(5) |
| | R²score | 0.9869(8) | 0.9623(7) | 0.9696(4) | -0.0166(3) | 0.9717(5) | 0.9720(5) | 0.2339(3) | 0.2133(5) |
| m=18 n=6 | MAE | **0.0008(1)** | 0.0013(5) | **0.0010(1)** | 0.0174(7) | 0.0021(4) | 0.0018(3) | **0.0191(1)** | 0.0181(3) |
| | RMSE | 0.0030(4) | 0.0065(5) | 0.0064(7) | 0.0365(7) | 0.0060(7) | 0.0059(7) | **0.0293(1)** | 0.0290(3) |
| | R²score | **0.9924(1)** | 0.9631(5) | 0.9639(7) | -0.1714(8) | 0.9685(7) | 0.9690(7) | **0.2465(1)** | 0.2648(3) |
| m=24 n=6 | MAE | **0.0008(1)** | 0.0012(2) | **0.0010(1)** | 0.0167(3) | 0.0021(4) | 0.0019(5) | **0.0191(1)** | **0.0175(1)** |
| | RMSE | **0.0029(1)** | 0.0056(2) | 0.0059(4) | 0.0359(5) | 0.0062(8) | 0.0062(8) | 0.0301(5) | **0.0276(1)** |
| | R²score | 0.9923(4) | 0.9725(2) | 0.9694(5) | -0.1410(5) | 0.9660(8) | 0.9664(8) | 0.1980(7) | **0.3269(1)** |
| Underlined indicator | | 3 | 3 | 3 | 0 | 0 | 2 | 0 | 0 |



**Table 6: Testing result of mode (5) Expanding-multi-R and multi-D.**

| Mode | Evaluation Indicator | Regression methods | | | | | | | |
|---|---|---|---|---|---|---|---|---|---|
| | | LR | TR | RFR | KNN | RR | KRR | LaR | ETN |
| m=6(3:3) n=3 | MAE | 0.0011(8) | 0.0014(6) | 0.0010(2) | 0.0168(1) | 0.0024(5) | 0.0024(8) | 0.0199(7) | 0.0191(7) |
| | RMSE | 0.0046(8) | 0.0076(6) | **0.0053(2)** | 0.0336(1) | 0.0061(8) | 0.0061(8) | 0.0308(7) | 0.0314(7) |
| | R²score | 0.9826(8) | 0.9528(5) | **0.9766(2)** | 0.0659(2) | 0.9697(6) | 0.9698(6) | 0.2183(7) | 0.1877(7) |
| m=12(9:3) n=3 | MAE | 0.0010(6) | 0.0013(3) | **0.0010(2)** | 0.0176(5) | 0.0024(5) | 0.0023(4) | 0.0197(5) | 0.0189(5) |
| | RMSE | 0.0039(4) | 0.0077(7) | 0.0061(3) | 0.0351(3) | 0.0060(4) | 0.0059(3) | 0.0303(5) | 0.0307(5) |
| | R²score | 0.9870(5) | 0.9506(7) | 0.9686(3) | -0.0358(3) | 0.9701(4) | 0.9702(4) | 0.2256(5) | 0.2040(5) |
| m=18(12:6) n=3 | MAE | 0.0009(4) | **0.0012(2)** | 0.0011(4) | 0.0181(7) | 0.0023(4) | 0.0023(4) | 0.0194(3) | 0.0185(3) |
| | RMSE | 0.0039(4) | 0.0061(2) | 0.0063(4) | 0.0367(7) | 0.0059(3) | 0.0059(3) | 0.0299(3) | 0.0299(3) |
| | R²score | 0.9871(4) | 0.9687(2) | 0.9665(5) | -0.1485(7) | 0.9704(3) | 0.9704(3) | 0.2382(3) | 0.2373(3) |
| m=24(18:6) n=3 | MAE | **0.0008(2)** | 0.0016(8) | 0.0011(4) | 0.0171(3) | 0.0020(2) | 0.0019(2) | **0.0191(1)** | **0.0179(1)** |
| | RMSE | **0.0029(2)** | 0.0141(8) | 0.0073(8) | 0.0358(5) | **0.0054(2)** | **0.0054(2)** | **0.0293(1)** | **0.0284(1)** |
| | R²score | **0.9924(2)** | 0.8246(8) | 0.9532(8) | -0.1241(5) | **0.9742(2)** | **0.9744(2)** | **0.2474(1)** | **0.2947(1)** |
| m=6(3:3) n=6 | MAE | **0.0000(1)** | **0.0001(1)** | **0.0001(1)** | **0.0168(1)** | **0.0005(1)** | **0.0005(1)** | 0.0199(7) | 0.0191(7) |
| | RMSE | **0.0000(1)** | **0.0007(1)** | **0.0005(1)** | **0.0336(1)** | **0.0025(1)** | **0.0025(1)** | 0.0308(7) | 0.0314(7) |
| | R²score | **1.0000(1)** | **0.9996(1)** | **0.9998(1)** | **0.0661(1)** | **0.9948(1)** | **0.9948(1)** | 0.2183(7) | 0.1877(7) |
| m=12(9:3) n=6 | MAE | 0.0010(6) | 0.0013(3) | 0.0011(4) | 0.0176(5) | 0.0024(5) | 0.0023(4) | 0.0197(5) | 0.0189(5) |
| | RMSE | 0.0040(7) | 0.0076(5) | 0.0063(4) | 0.0351(3) | 0.0060(4) | 0.0060(6) | 0.0303(5) | 0.0307(5) |
| | R²score | 0.9868(7) | 0.9510(6) | 0.9666(4) | -0.0358(3) | 0.9699(5) | 0.9701(5) | 0.2256(5) | 0.2040(5) |
| m=18(12:6) n=6 | MAE | 0.0009(4) | 0.0013(3) | 0.0011(4) | 0.0181(7) | 0.0024(5) | 0.0023(4) | 0.0194(3) | 0.0185(3) |
| | RMSE | 0.0039(4) | 0.0064(3) | 0.0065(6) | 0.0367(7) | 0.0060(4) | 0.0060(6) | 0.0299(3) | 0.0299(3) |
| | R²score | 0.9870(5) | 0.9648(3) | 0.9635(6) | -0.1485(7) | 0.9695(7) | 0.9696(7) | 0.2382(3) | 0.2373(3) |
| m=24(18:6) n=6 | MAE | 0.0008(2) | 0.0014(6) | 0.0011(4) | 0.0171(3) | 0.0022(3) | 0.0021(3) | **0.0191(1)** | **0.0179(1)** |
| | RMSE | 0.0030(3) | 0.0070(4) | 0.0071(7) | 0.0358(5) | 0.0060(4) | 0.0059(3) | **0.0293(1)** | **0.0284(1)** |
| | R²score | 0.9924(2) | 0.9566(4) | 0.9557(7) | -0.1241(5) | 0.9687(8) | 0.9690(8) | **0.2474(1)** | **0.2947(1)** |
| Underlined indicator | | 6 | 4 | 6 | 0 | 2 | 2 | 0 | 0 |

Among the five modes, mode (1) adopts the traditional waterlogging forecasting style. It only uses rainfall data to predict waterlogging accumulation and gets the worst testing result. In this mode, the larger m is, the more information the model learns and the better the testing performance is. When m changes from 6 to 24, the $R^2$ score of RR changes from 0.1209 to 0.3954, which is 3.27 times of the initial value. However, when m exceeds 24, the noise increases significantly. By comparing the predicted value with the actual value, the results of the eight kinds of regression methods all have large noise when the actual waterlogging depth is 0. Even though KRR and LR can achieve good trend prediction at the peak, there are still large noise fluctuations in most of the time (**Figure 7**). This phenomenon may be caused by the lack of historical waterlogging time series input, so the noise suppression is not good. In view of the poor prediction results, mode (1) is not included in the discussion of optimal indicators, and only the results from mode (2) to mode (5) are counted.




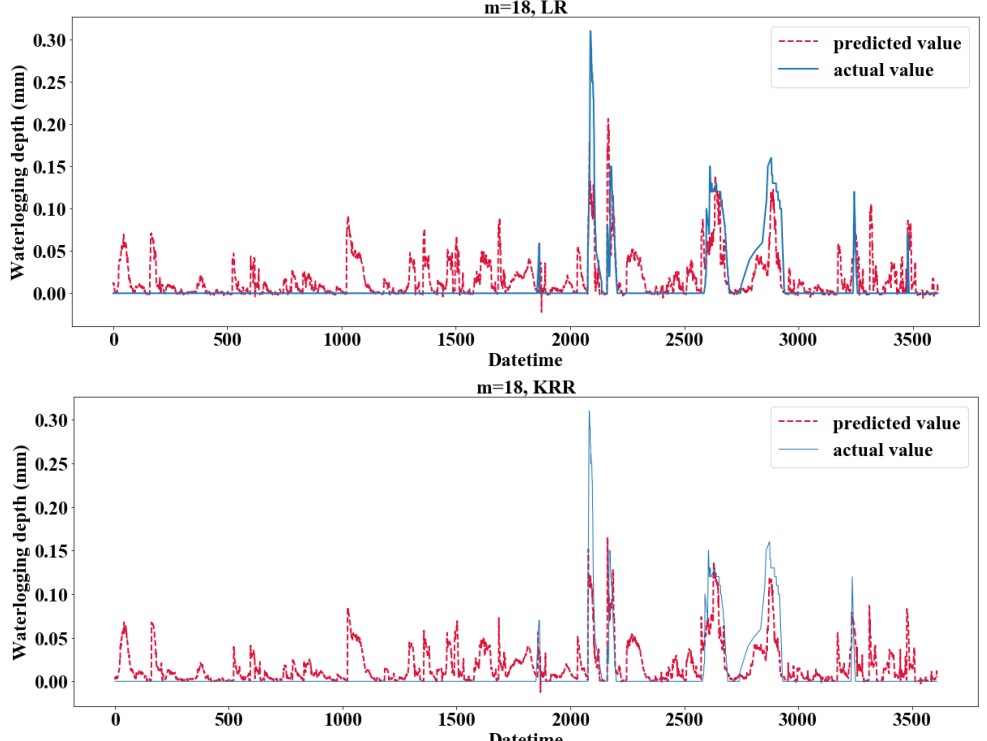

**Figure 7: Testing result using LR and KRR method when m=18.**

In mode (2), the prediction performance of LR and TR becomes better as m increases, which is the same as that of mode (1). However, RFR, RR, KRR are not sensitive to parameter changes. The structure of mode (3) and mode (2) is just opposite. It is worth noting that in mode (3), the larger the parameter n is, the better the model may not be. For example, the optimal results of RFR, RR and KRR are obtained when n=6, which the $R^2$score of the three exceeds 0.977, indicating that the early information of waterlogging depth is not helpful to the prediction of waterlogging depth in the future, and may cause some

interference. For LR, with the same number of parameters, the result of mode (3) is better than that of mode (2), and MAE is 0.0006. The main reason is that mode (3) extracted more historical waterlogging depth information, which changes by a continuous process in short-term prediction, the linear model can better estimate the output of the next time. However, this does not mean that the model has the best performance, because it contains insufficient rainfall information and may not perform well in the practical application of prediction.

Mode (4) coupled multiple rainfall and waterlogging inputs. Overall, the results of mode (4) are better than that of mode (2) and mode (3) with one-dimensional input (m=1 or n=1). In this mode, the TR and RFR methods achieved the best testing results. TR achieved **0.9778** $R^2$score and 0.0050 RMSE (m=24, n=3). The $R^2$score of RFR reached **0.9803** and RMSE reached **0.0049** (m=6, n=3).

In mode (5), when parameters are adjusted to m=6(3:3) and n=6, LR evaluation indicator (MAE=0, RMSE=0 and $R^2$score=1)

is abnormal. This problem also exists in RFR, TR, etc. The main reason is that the prediction label has been included in the



input time series, so the result of m=6(3:3), n=6 in mode (5) should be removed in the discussion. Based on the performance of mode (5) on the test set, the performance of mode (5) will get better after the predicted time is extended. The reason is that mode (5) extends the rainfall input and considers the influence of future rainfall. Especially, the rainfall model with short duration and high intensity is more suitable for this mode. There is also a special case in mode (3) when m=0, only waterlogging

data is used to predict future water changes. This mode is not listed separately in this paper, because with the increase of prediction time, the previous information of waterlogging can no longer accurately predict the trend of subsequent value, and the model performance will decline rapidly. As shown in **Figure 8**, (a) only 30 minutes of waterlogging data were used for prediction, and (b) 30 minutes of waterlogging with 90 minutes of extended rainfall were used as input. With the increase of prediction time, the difference of prediction performance between them increases gradually. In the prediction of 40minutes

waterlogging, $R^2$score of (b: 0.6841) is 2.4 times that of (a: 0.2847) when TR method is applied. Using the RFR method, $R^2$score and MAE of (b: $R^2$score=0.7428, MAE=0.0044) also significantly exceeds (a: $R^2$score=0.6488, MAE=0.0053). Especially for the prediction of medium-high value, in the case of high value (0.30m) and medium value (0.13m), the prediction results of (a) are 0.82m and 0.73m, and the large error leads to poor accuracy in the prediction of medium-large scale waterlogging.

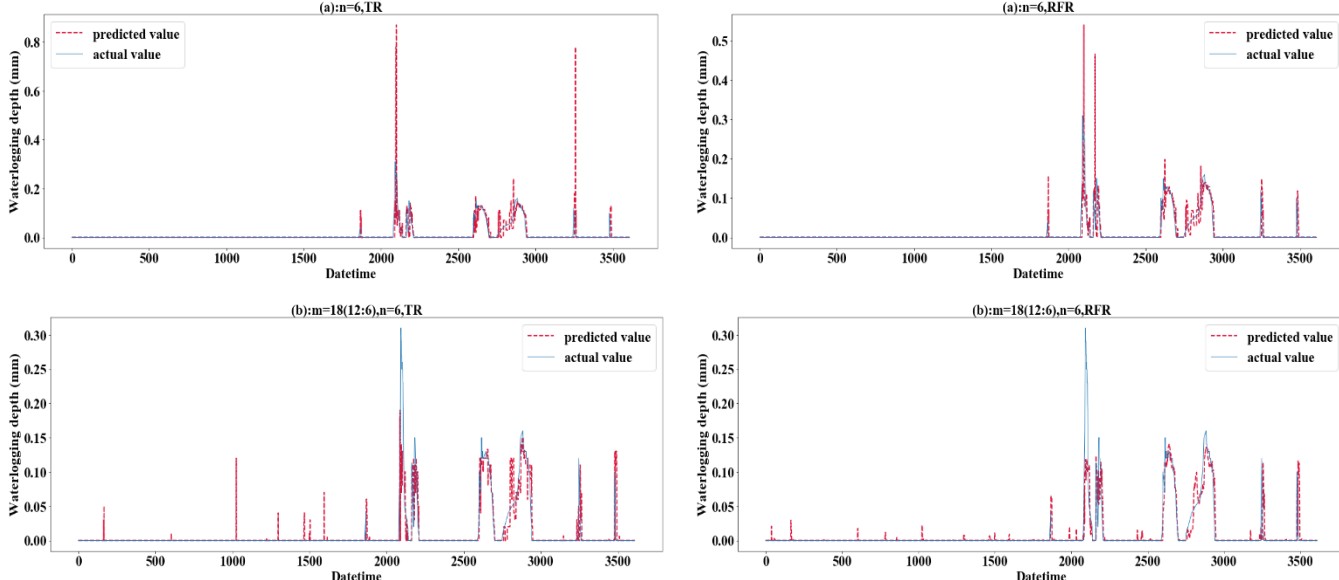

**Figure 8: Testing result of mode (5) and only-D prediction using TR and RFR (when both n=6).**





Figure 9: Testing result using different method when m=24, n=3. (a) for KRR, (b) for LR, (c) for RFR, (d) for TR.





From the perspective of comparison of regression methods, the performance of LR, TR, RFR, RR and KRR are relatively better, which is reflected in the strong generalization ability of the model (**Figure 9**). KRR, as Ridge Regression with kernel function added, is more suitable for high-dimensional data. In this study, it shows a slightly stronger regression performance

than RR. It can be seen from the comparison (**Figure 9(a)(b)**) that LR and KRR have strong prediction ability for high values, but poor noise suppression for low values and 0 values, and the model fluctuates constantly around the x axis. The prediction of RFR for the highest value is insufficient, but the prediction performance for other high values is better. Its noise control for low values is better. TR has the best noise control effect for 0 value, but the curve is not smooth or ladder shape at high value. Of course, this is related to the principle of the algorithm. When applying RFR, selecting the parameter n_estimators which is

equal to 100 can solve the problem of TR (**Figure 9(c)**). LR, RFR, KRR, TR show strong fitting ability in the training set (TR has the MAE:0.0000 RMSE:0.0000 score:1.0000) (**Figure 10**) in, KNN and ETN show relatively poor fitting ability. KNN, LaR and ETN have weak ability of fitting and generalization, which are not suitable for regression prediction of such data (**Figure 11**). KNN methods have negative $R^2$score in mode (1), mode (2), mode (4) and mode (5). For most data sets with eigenvalues of 0, the prediction performance is often poor, which can explain why the results of KNN method are the worst

(**Figure 11(a)**). LaR and ETN are mode-insensitive and have the same results in mode (1) and mode (2). However, within each mode, as m and n change, the results will be different. The poor results of the LaR method (**Figure 11(c)**) may be that the method is suitable for multi-variable model, and the variables are selected by adjusting the λ value to change the compression variable coefficient, but there are fewer variables in this study. Similarly, considering that ETN (**Figure 11(b)**) works well when many features are interconnected, while this model has a small number of features, the model with similar basic principles

to Lasso Regression also has poor performance.





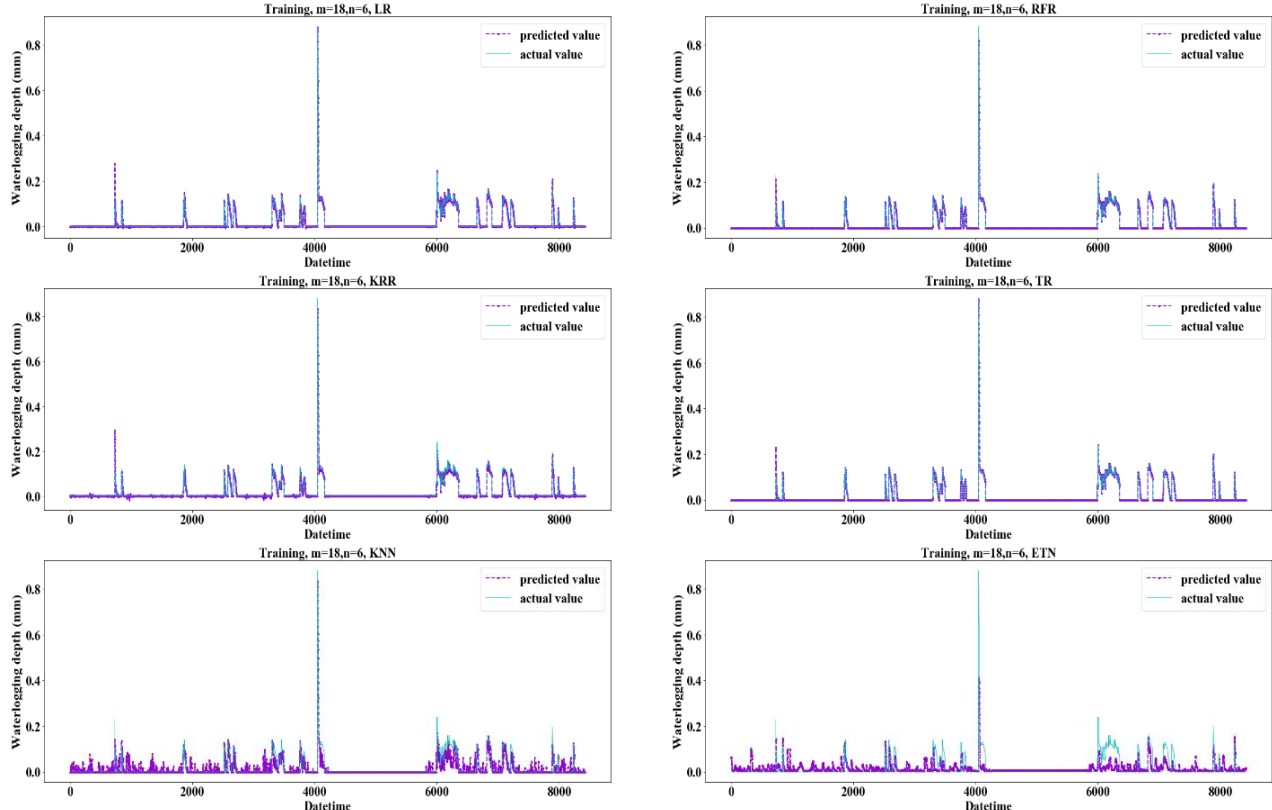

**Figure 10: Fitting performance of different regression methods on training set when m=18, n=6.**


**Figure 11: Performance of KNN, ETN and LaR method on testing set when m=18, n=6. (a) for KNN, (b) for ETN, (c) for Las.**

What needs special attention is that, according to the evaluation indicator, LR seems to have achieved good prediction results in all the five modes. However, the factor that cannot be ignored is that the original waterlogging depth data is sparse and uneven, which must be resampling interpolation processing. In the interpolation process, we used linear interpolation method, which may cause the result to be higher than the actual value when using LR. Therefore, it is necessary to go through actual





value test to judge whether LR method is really applicable to prediction. As an improved LR method, RR adds L2 regularization and modified cost function on the basis of linear regression, so LR and RR have the same results in mode (1).

### 4.5.2 Prediction verification based on actual value

Actual value verification takes a subset of the testing set, so the first 85% of the full data set is selected as the training set, which can increase the number of training samples and improve the training ability of the model. The time series from 2020/5/26 13:00 to 2020/5/26 17:30 was selected, lasting 4.5h (**Figure 12**), covering the complete process of waterlogging fluctuation. In this way, the prediction ability of the model for the change trend of waterlogging depth can be verified. The changes of rainfall and waterlogging in this period are shown in the figure. During verification, the time series was divided

into 6 groups on average. Starting from the first moment of each group, the waterlogging depth changes of 30 minutes were predicted by 6 steps forward in each period.

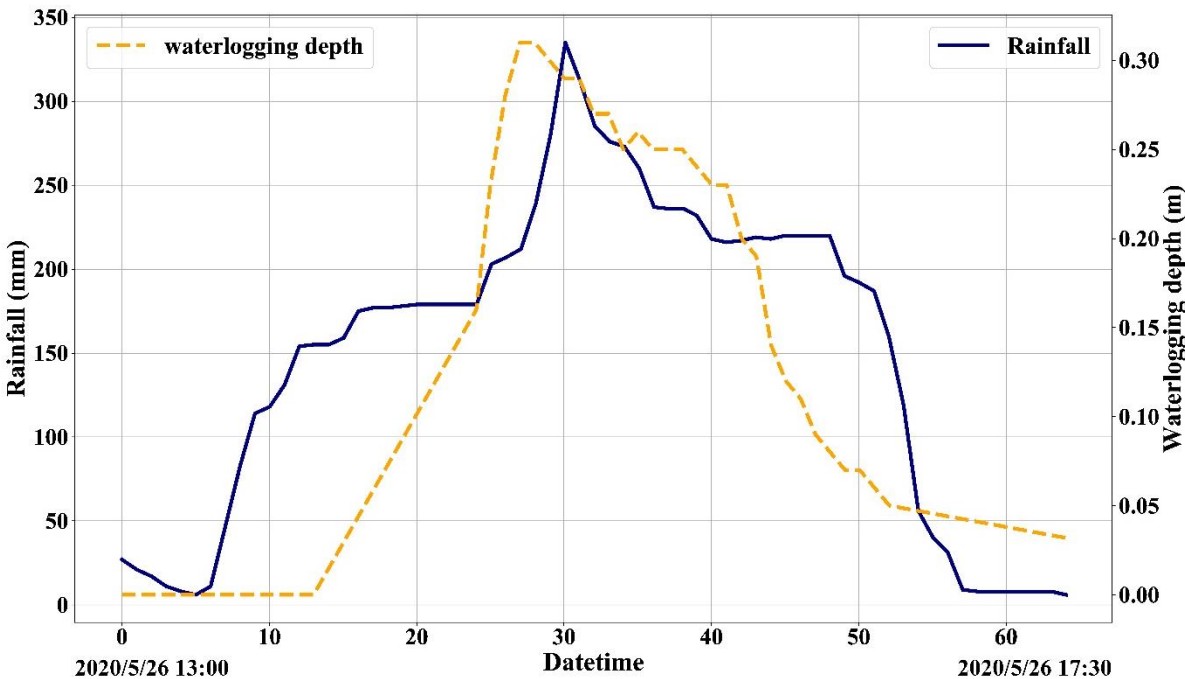

**Figure 12: Actual value verification time series from 2020/5/26 13:00 to 17:30.**

In order to highlight the prediction performance of each strategy, we set s=6 for 30-minute-prediction. That is a long forecast

for real-time water levels. In practical application, 15 minutes to 20 minutes prediction may be the common prediction, and 20 minutes prediction has fully met the requirement of releasing early warning information in advance and dispatching the nearest traffic and fire personnel to the scene for disposal (Georgiadou et al., 2010). We will make evaluation from multiple indicators such as Absolute Error (**AE**), Absolute Percent Error (**APE**) and **Time Cost**. Mode (5) with m=18(12:6) and n=6 was selected as the basic parameters of the model to evaluate the model performance under each strategy. **Figure 13** represents the isometric





segmentation of the verification data set with different strategies. The first 30 minutes of the time series are taken to show the
        curves between the actual values in each group and the predicted values for each method. As can be seen from **Figure 13**,
        KNN, LaR and ETN, perform poorly in the application of prediction. They have too high AE and APE, and the predicted value
        was far from the actual value. Therefore, we will not discuss these three poor methods in the analysis of results. **Figure 14**
        reflects Absolute Error of predicted value and actual value of each strategy. We set a tolerance of 0.02m to exclude reasonable
error. APE reflects the accuracy of the model, but at low values it does not independently reflect the performance of the model
        because even small errors (< 0.01m) can cause APE to rise (**Figure 15**). Therefore, the mean value of AE and APE (MAE and
        MAPE) was used to evaluate the model accuracy, and the variance of AE and APE (V-AE and V-APE) was used to evaluate
        the robustness. Since real-time sensor data is used in prediction, continuous calculation is required to update the model. Time
        cost is adopted as an indicator to evaluate the continuous computing capability of the model.



**Figure 13: Predicted value and actual value of different strategies. (The x-axis represents predicted steps within each group)**




**Figure 14: Absolute Error (AE) of different strategies. (The x-axis represents predicted steps within each group)**








**Figure 15: Absolute Percent Error (APE) of different strategies. (The x-axis represents predicted steps within each group)**




As seen in **Table 10**, MAE of Rec strategy is 0.0153m, which is better than 0.0184m of SOC and 0.0165m of MO. Of course, the MAE of the three is within the tolerance (0.02m). Under the Rec strategy, the model replaces the last data point of the input vector with the predicted value every step. The reason for the optimal REC result is that the fluctuation of the waterlogging is

basically a monotone increasing or decreasing process, and generally there is no fluctuation in a short period of time. In addition, the input vector is long enough to minimize the cumulative error caused by the predicted value deviation. The application of recursive algorithm can better continue the prediction trend of the model, so the MAE result of Rec is the best.

Rec and MO have lower MAPE, corresponding accuracy can reach 86.1% and 85.7% The MAPE of SOC was 0.186 and the accuracy was only 81.4%. The main reason for this phenomenon is the working mechanism of SOC, which adopts six separate

models to combine the prediction results into the final prediction vector. When s=1, the prediction strategy is the same as MO. However, with the increase of S, when predicting the s step, there will be a loss of $y_{t+1}, y_{t+2}, \cdots, y_{t+s-1}$ in the middle, resulting in a large error in the following part. Meanwhile, the V-AE and V-APE of SOC strategy are the worst, which represents poor performance and robustness of the model.

MO has the best V-APE (0.01036), indicating that the accuracy of this strategy has the smallest fluctuation, and the model has

the best robustness. The V-AE (0.00026) of Rec was similar to that of MO (0.00022), but the V-APE of Rec was 69% larger than that of MO. The reason is that APE fluctuated greatly when TR and RFR were applied under Rec strategy, and V-APE was 3.05 times and 2.49 times of MO strategy respectively. Therefore, we infer that under the same conditions, the decision tree algorithm has smaller fluctuation when applied to multidimensional output and can obtain better robustness.

**Table 7: Model performance using Rec strategy.**

| P | EI | LR | TR | RFR | RR | KRR | P | EI | LR | TR | RFR | RR | KRR |
|---|---|---|---|---|---|---|---|---|---|---|---|---|---|
| 1 | MAE | 0.0284 | 0.0223 | 0.0139 | 0.0443 | 0.0436 | 4 | MAE | 0.0112 | 0.0076 | 0.0058 | 0.0045 | 0.0042 |
|  | MAPE | 0.2109 | 0.1661 | 0.1062 | 0.3469 | 0.3419 |  | MAPE | 0.2063 | 0.1388 | 0.1144 | 0.0829 | 0.0780 |
|  | V (AE) | 0.0003 | 0.0002 | 0.0001 | 0.0003 | 0.0002 |  | V (AE) | 0.0001 | 0.0001 | 0.0000 | 0.0001 | 0.0000 |
|  | V (APE) | 0.0085 | 0.0079 | 0.0037 | 0.0036 | 0.0034 |  | V (APE) | 0.0085 | 0.0042 | 0.0091 | 0.0021 | 0.0020 |
| 2 | MAE | 0.0264 | 0.0317 | 0.0119 | 0.0060 | 0.0061 | 5 | MAE | 0.0003 | 0.0035 | 0.0025 | 0.0010 | 0.0002 |
|  | MAPE | 0.0976 | 0.1179 | 0.0446 | 0.0221 | 0.0222 |  | MAPE | 0.0096 | 0.0999 | 0.0705 | 0.0291 | 0.0069 |
|  | V (AE) | 0.0001 | 0.0003 | 0.0001 | 0.0000 | 0.0000 |  | V(AE) | 0.0001 | 0.0001 | 0.0000 | 0.0000 | 0.0000 |
|  | V (APE) | 0.0020 | 0.0048 | 0.0012 | 0.0001 | 0.0001 |  | V(APE) | 0.0001 | 0.0033 | 0.0012 | 0.0002 | 0.0000 |
| 3 | MAE | 0.0233 | 0.0483 | 0.0553 | 0.0195 | 0.0196 | 6 | MAE | 0.0009 | 0.0028 | 0.0064 | 0.0034 | 0.0025 |
|  | MAPE | 0.1398 | 0.2819 | 0.3151 | 0.1219 | 0.1219 |  | MAPE | 0.0486 | 0.1446 | 0.3544 | 0.1776 | 0.1319 |
|  | V (AE) | 0.0005 | 0.0014 | 0.0013 | 0.0005 | 0.0005 |  | V (AE) | 0.0001 | 0.0001 | 0.0000 | 0.0000 | 0.0000 |
|  | V (APE) | 0.0266 | 0.0789 | 0.0756 | 0.0273 | 0.0272 |  | V (APE) | 0.0015 | 0.0155 | 0.1017 | 0.0100 | 0.0059 |


**Table 8: Model performance using SOC strategy.**

| P | EI | LR | TR | RFR | RR | KRR | P | EI | LR | TR | RFR | RR | KRR |
|---|---|---|---|---|---|---|---|---|---|---|---|---|---|
| 1 | MAE | 0.0349 | 0.0515 | 0.0229 | 0.0554 | 0.0549 | 4 | MAE | 0.0060 | 0.0047 | 0.0043 | 0.0032 | 0.0037 |
|  | MAPE | 0.2608 | 0.3846 | 0.1878 | 0.4263 | 0.4228 |  | MAPE | 0.1261 | 0.1320 | 0.1488 | 0.1003 | 0.1054 |
|  | V (AE) | 0.0005 | 0.0008 | 0.0002 | 0.0006 | 0.0006 |  | V (AE) | 0.0001 | 0.0001 | 0.0001 | 0.0001 | 0.0001 |
|  | V (APE) | 0.0179 | 0.0289 | 0.0121 | 0.0129 | 0.0128 |  | V (APE) | 0.0048 | 0.0257 | 0.0237 | 0.0098 | 0.0093 |
| 2 | MAE | 0.0284 | 0.0237 | 0.0274 | 0.0176 | 0.0178 | 5 | MAE | 0.0015 | 0.0302 | 0.0071 | 0.0018 | 0.0008 |
|  | MAPE | 0.1103 | 0.1103 | 0.0847 | 0.0697 | 0.0702 |  | MAPE | 0.0574 | 0.7403 | 0.1732 | 0.0659 | 0.0407 |
|  | V (AE) | 0.0001 | 0.0013 | 0.0002 | 0.0001 | 0.0001 |  | V(AE) | 0.0002 | 0.0010 | 0.0000 | 0.0001 | 0.0000 |
|  | V (APE) | 0.0028 | 0.0191 | 0.0029 | 0.0021 | 0.0022 |  | V(APE) | 0.0026 | 1.1122 | 0.0350 | 0.0026 | 0.0023 |
| 3 | MAE | 0.0276 | 0.0400 | 0.0334 | 0.0200 | 0.0202 | 6 | MAE | 0.0020 | 0.0039 | 0.0022 | 0.0034 | 0.0022 |
|  | MAPE | 0.1804 | 0.2680 | 0.2216 | 0.1423 | 0.1434 |  | MAPE | 0.1257 | 0.1872 | 0.1612 | 0.1939 | 0.1399 |
|  | V (AE) | 0.0006 | 0.0013 | 0.0006 | 0.0005 | 0.0005 |  | V (AE) | 0.0000 | 0.0000 | 0.0000 | 0.0000 | 0.0000 |
|  | V (APE) | 0.0352 | 0.0633 | 0.0333 | 0.0297 | 0.0302 |  | V (APE) | 0.0110 | 0.0178 | 0.0144 | 0.0138 | 0.0099 |




**Table 9: Model performance using MO strategy.**

| P | EI | LR | TR | RFR | RR | KRR | P | EI | LR | TR | RFR | RR | KRR |
|---|---|---|---|---|---|---|---|---|---|---|---|---|---|
| 1 | MAE | 0.0349 | 0.0536 | 0.0284 | 0.0554 | 0.0549 | 4 | MAE | 0.0060 | 0.0074 | 0.0045 | 0.0032 | 0.0037 |
|  | MAPE | 0.2564 | 0.4092 | 0.2119 | 0.4257 | 0.4217 |  | MAPE | 0.1077 | 0.1353 | 0.0833 | 0.0561 | 0.0659 |
|  | V (AE) | 0.0005 | 0.0006 | 0.0003 | 0.0006 | 0.0006 |  | V (AE) | 0.0001 | 0.0001 | 0.0001 | 0.0001 | 0.0001 |
|  | V (APE) | 0.0168 | 0.0155 | 0.0082 | 0.0128 | 0.0125 |  | V (APE) | 0.0030 | 0.0065 | 0.0039 | 0.0014 | 0.0020 |
| 2 | MAE | 0.0284 | 0.0100 | 0.0230 | 0.0176 | 0.0178 | 5 | MAE | 0.0015 | 0.0027 | 0.0071 | 0.0018 | 0.0008 |
|  | MAPE | 0.1052 | 0.0370 | 0.0834 | 0.0659 | 0.0664 |  | MAPE | 0.0430 | 0.0779 | 0.2127 | 0.0518 | 0.0223 |
|  | V (AE) | 0.0001 | 0.0001 | 0.0001 | 0.0001 | 0.0001 |  | V(AE) | 0.0002 | 0.0002 | 0.0000 | 0.0001 | 0.0000 |
|  | V (APE) | 0.0025 | 0.0010 | 0.0009 | 0.0019 | 0.0020 |  | V(APE) | 0.0009 | 0.0016 | 0.0392 | 0.0009 | 0.0002 |
| 3 | MAE | 0.0276 | 0.0183 | 0.0328 | 0.0200 | 0.0202 | 6 | MAE | 0.0020 | 0.0039 | 0.0011 | 0.0034 | 0.0022 |
|  | MAPE | 0.1641 | 0.1075 | 0.1844 | 0.1243 | 0.1256 |  | MAPE | 0.1065 | 0.2029 | 0.0544 | 0.1771 | 0.1135 |
|  | V (AE) | 0.0006 | 0.0003 | 0.0004 | 0.0005 | 0.0005 |  | V (AE) | 0.0003 | 0.0003 | 0.0000 | 0.0001 | 0.0001 |
|  | V (APE) | 0.0334 | 0.0145 | 0.0231 | 0.0280 | 0.0285 |  | V (APE) | 0.0066 | 0.0273 | 0.0019 | 0.0096 | 0.0037 |

Considering the time cost, the model must be updated within the 5 minutes interval for data prediction. This study used tower workstation with 8 core Intel(R) Xeon(R) W-2123 3.60 GHz CPU, 64.0 GB RAM and NIVDIA Quadro RTX4000 GPU. The

average time cost of Rec and SOC is about 2.54 times that of MO. Under the MO strategy, the output form of the model is a six-dimension-vector. The advantage of this strategy is that it only needs to perform one training and prediction process, and does not need recursion or multiple model coupling, so it is more convenient to use. Therefore, the calculation time of each regression method is greatly reduced. The time cost of KRR was shortened from 240.39s to 83.28s. It should be noted that although KRR algorithm has good model performance and robustness, its time cost is too high, reaching 241.28s under Rec

strategy, which is difficult to meet the requirements of update calculation within 5 minutes. LR, TR and RR can all be updated within three seconds due to their simple structures. RFR has a high time cost because it has to traverse all trees. However, it can meet the requirement of update time (only 65.13s under MO strategy).

**Table 10: Model performance statistics with different strategies applied**

| Strategy | ACC-30min | Average indicator | | Evaluation indicator | Regression methods | | | | |
|---|---|---|---|---|---|---|---|---|---|
|  |  |  |  |  | LR | TR | RFR | RR | KRR |
| Rec | 86.17% | Avg (MAE) | 0.01526 | MAE-Rec | 0.0905 | 0.0194 | 0.0160 | 0.0131 | 0.0127 |
|  |  | Avg (MAPE) | 0.13834 | MAPE-Rec | 0.1188 | 0.1582 | 0.1675 | 0.1301 | 0.1171 |
|  |  | Avg (VAE) | 0.00022 | V (AE)-Rec | 0.0002 | 0.0003 | 0.0003 | 0.0002 | 0.0001 |
|  |  | Avg (VAPE) | 0.0175 | V (APE)-Rec | 0.0079 | 0.0339 | 0.0321 | 0.0072 | 0.0064 |
|  |  | Avg (TimeC) | 75.914s | Time cost-Rec(s) | 0.19 | 2.38 | 135.60 | 0.12 | 241.28 |
| SOC | 81.40% | Avg (MAE) | 0.01842 | MAE-SOC | 0.0167 | 0.0257 | **0.0162** | 0.0169 | 0.0166 |
|  |  | Avg (MAPE) | 0.18604 | MAPE-SOC | **0.1435** | 0.3037 | 0.1629 | 0.1664 | 0.1537 |
|  |  | Avg (VAE) | 0.00034 | V (AE)-SOC | 0.0003 | 0.0008 | **0.0002** | **0.0002** | **0.0002** |
|  |  | Avg (VAPE) | 0.05334 | V (APE)-SOC | 0.0124 | 0.2112 | 0.0202 | 0.0118 | **0.0111** |
|  |  | Avg (TimeC) | 77.194s | Time cost-SOC(s) | 0.20 | 2.40 | 142.84 | 0.14 | 240.39 |
| MO | 85.69% | Avg (MAE) | 0.01648 | MAE-MO | 0.0167 | 0.0160 | 0.0162 | 0.0169 | 0.0166 |
|  |  | Avg (MAPE) | 0.14332 | MAPE-MO | 0.1305 | 0.1616 | 0.1384 | 0.1502 | 0.1359 |
|  |  | Avg (VAE) | 0.00026 | V (AE)-MO | 0.0003 | 0.0003 | 0.0002 | 0.0003 | 0.0002 |
|  |  | Avg (VAPE) | 0.01036 | V (APE)-MO | 0.0105 | 0.0111 | 0.0129 | 0.0091 | 0.0082 |
|  |  | Avg (TimeC) | 29.94s | **Time cost-MO(s)** | **0.09** | **1.14** | **65.13** | **0.06** | **83.28** |





In conclusion, LR, TR, RFR, RR and KRR are superior to other methods, which is consistent with the results on the testing

set. Models using Rec or MO strategies have better performance and robustness, with average accuracy more than 85% for

predicting waterlogging depth in the next 30 minutes. For short-term prediction, such as 15 minutes prediction, the accuracy

can reach 93%, and the robustness of the model will be further improved. As can be seen from **Figure 13** to **Figure 15**, when

s = 3, the prediction curves of RFR, LR, KRR and other methods basically match actual value, and AE and APE of each group

are almost within tolerance.

## 5 Conclusions and Prospects

### 5.1 Conclusions

The prediction and early warning of urban rainstorm and waterlogging disaster has always been the key problem concerned

by all city safety managers. It is difficult to predict the rapid water level rise caused by short-term heavy rainfall in advance.

The model with rainfall and waterlogging depth can be built by using the historical water level information and weather data

of the sensor for model training. Combining the current water level information with the short-term weather information in the

future, the fluctuation of waterlogging can be predicted accurately in a short time. Recursive strategy is considered to be a poor

prediction strategy in Ben Taieb's research (Ben Taieb et al., 2012), because his data presents periodic characteristics. In this

study, physical characteristics of waterlogging determine that water level change is generally a process of monotonically

increasing or decreasing, so Rec can also have a good performance on the prediction of non-periodic data with obvious trends.

MO is a strategy that considers both accuracy and computational efficiency. In the long-term forecast, the rainfall factor makes

up for the lack of information about the variable of water accumulation, which can improve the model performance, but it is

not obvious in the short-term forecasting.

Waterlogging caused by rainstorms usually accumulates in low-lying areas of the city, such as poorly drained blocks and roads,

underpass tunnels, bridge culverts, municipal plumbing manholes, and underground shopping malls or parking lots. Accurate

prediction of waterlogging before it reaches the warning threshold can get more time for emergency decision-making and

disaster response. Government emergency departments can timely issue warning information to the public, notify traffic

management departments to rush to the scene to block the relevant roads, culverts, tunnels, etc. Waterlogging sensors are

usually located in places where waterlogging often occurs. The Shenzhen municipal government has been committed to

eliminating these risk points through the renovation of municipal facilities, but this is a complex and lengthy project. Effective

forecasting and monitoring will help minimize casualties and property losses until waterlogging risk points are completely

eliminated.

In the study of Huang et al., the depth of waterlogging was calculated and predicted by recognizing the submersion of

automobile tires in waterlogging, and Shenzhen was taken as a case (Huang et al., 2020). The results of this method can be

disturbed by many factors, such as tire model size, water surface fluctuation and visibility. However, the research in this paper

is based on real waterlogging depth data, without some inevitable errors of other methods, so the accuracy is relatively high.



## 5.2 Prospects

- Due to the limited number of waterlogging sensors, it cannot cover a large area, so the geographical information richness of the location point is insufficient. From this perspective, spatial flood prediction was not involved in this study, as we did not study prediction models used to identify the location of floods. In fact, we were concerned only with the lead time for an identified site. In the future, increasing the number of sensors can improve the geographic information of waterlogging point location, including more DEM, slope, positive and negative terrain, infiltration rate and other information. This kind of model can be extended to the spatial dimension for prediction. Through grid analysis, all position points in the study area can be traversed and waterlogging risk map can be drawn.

- Due to the working mechanism of the waterlogging depth sensor, the sampling rate is not constant. In order to unify the dimensions of model input data, oversampling and interpolation methods must be used to complete the data in the period of sparse data. Although the missing data is largely due to the water level of 0, individual data may still be missing. Different interpolation methods have different mechanisms for data completion, so there is some error between interpolation and actual value (Agerberg, 2020). Although through the comparison of interpolation methods, the optimal interpolation method has been selected to minimize the error, but it is still not as reliable and accurate as improving the sampling rate of the sensor. When determining the interpolation method, the error between interpolation and the actual value can be further corrected by calculating the derivative and the second derivative, but it is not considered in this paper.

- At present, this paper mainly studies the relationship between rainfall and the depth of waterlogging but the waterlogging is also affected by other meteorological characteristics. For example, differences in evaporation and infiltration of surface runoff caused by seasonal factors should be considered, and input of other meteorological data should be added to eliminate such seasonal differences. Relative humidity and air temperature are the most relevant factors. These two kinds of data can be added to enhance the training performance in future model improvement (Shozib & Rahman, 2021).

- Some regression algorithms, such as Bayesian Regression, GBRT Regression and AdaBoosting Regression, cannot carry out multidimensional output directly (Zhan et al., 2019). Considering the model performance of MO strategy is to be compared, the above algorithms are not selected. However, the prediction performance of Bayesian Regression under Rec and SOC strategy is pretty good. Therefore, the algorithm can also be included in the application of the framework.

Acknowledgements. This research was funded by National Key R&D Program of China (2018YFC0807000), Natural Science Foundation of China (71771113), and National Key R&D Program of China (2019YFC0810705). It was also partly supported by the High-level Special Funding of the Southern University of Science and Technology (grant nos. G02296302 and G02296402).

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
