# Peer review of "A Multi-strategy-mode-waterlogging-prediction Framework for Urban Flood Depth"

_Natural Hazards and Earth System Sciences, 2022_

## Author Comment (AC1)

Thank you very much for your insightful comments. Please find our responses as follows.

**Q1:**

In the literature review, the authors introduced Physically based models, Statistical methods and Data-driven models, but do not explore the connection between the previous study and the study.

**A1:** Current research on prediction and early warning of urban waterlogging disaster is mainly based on physically based models, statistical methods and data-driven models. The physically based models have the advantage of interpretability, but they have strong requirements on the data of urban underlying surface and rely on complicated calculation, and thus have difficulties in large-scale applications. Statistical methods have low requirements for urban underlying surface data, but the weight settings of each factor will greatly affect the final results. The framework proposed in this paper is based on the data-driven model, which takes the advantage of the machine learning methods and the availability of huge amount of sensor-generated data. Hossein et al. presented a flood simulation framework, which used a random forest classification model and a multilayer perceptron model to identify wet or dry nodes over the domain, then estimate river depth in wet nodes. The flood simulation framework cannot predict the depth of waterlogging in real time because it is based on different relative flow grades. Our framework is based on the data-driven model, which has fitted many years of real data and has a good performance in terms of prediction accuracy and calculation efficiency.

**Q2:**

The authors developed a multi-strategy-mode that can forecast the urban flood depth. The 3 strategies used by the authors are included in the 5 strategies commonly used in time series forecasting (such as Recursive, Direct, DirRec, MIMO, DIRMO). Please explain difference between multi-strategy-mode and the commonly used strategies.

**A2:** The prediction strategies used in this paper are indeed commonly used strategies in time series forecasting. However, in this paper we focus on the evaluation of these strategies with respect to the problem of urban waterlogging prediction to shed light on which strategy is more applicable and can provide more accurate results. For example, in the literature "A review and comparison of strategies for multi-step ahead time series forecasting based on the NN5 forecasting competition", Ben Taieb et al. compared the performance of some common prediction strategies for the prediction of daily cash money withdrawal amounts at ATM machines. In their paper, the experiment results showed that the multi-step strategy achieved the best performance, while the Rec strategy achieved the worst performance. However, in the physical process of the change of waterlogging depth, the curve is monotonous, and therefore the Rec strategy is more adaptable.

**Q3:**

The prediction accuracy of 81.6% is a result of one of three tried strategie (Rec), it is not a result of multiple strategies fusion.

**A3:** The 86.1% accuracy is the result of Rec prediction strategy. The multi-strategy here means that this framework includes the steps to select the optimal strategy, and the applicability of each prediction strategy is verified through multiple groups of experiments. The accuracy and principle of the optimal prediction strategy in the application of waterlogging prediction are explained. This is different from the concept of "coupling strategies" and does not mean that multiple strategies are combined into a new one. In future studies, we will consider coupling different strategies, modes

and algorithms to improve the framework.

**Q4:**

The paper said that "accuracy of predicting is superior to many data-driven prediction models for waterlogging depth", I hope the authors give examples and discuss further.

**A4:** In the literature, the accuracy of some data-driven models is below 85% in 30-minute prediction. For example, Jing H et al. proposed a novel approach to measure urban waterlogging depth by Mask R-CNN. its accuracy rate can reach 80.52%(video) and 81.38%(image). However, since the experiment setting is not the same, this conclusion may not be accurate. We will modify the manuscript to clear the confusion.

---

## Author Comment (AC2)

Dear Editors and Reviewers:

Thank you for your letter and for the reviewers' comments concerning our manuscript entitled "A Multi-strategy-mode-waterlogging-prediction Framework for Urban Flood Depth" (ID: nhess-2022-36). Those comments are all valuable and very helpful for revising and improving our paper, as well as the important guiding significance to our research. We have studied comments carefully and have made the correction which we hope meet with approval. The main corrections in the manuscript and the responses to the reviewer's comments are as flowing:

Comment #1: **A flowchart is necessary for the case study as well, as there have been many details regarding data extraction, preprocessing and comparison between methods, etc. in the case study.**

**Reply:** Your comment is constructive, and we believe that a flowchart would more clearly reflect the steps of the case study and allow for a more progressive approach to the results and conclusions. We have modified the case study sections according to the process steps in the framework and expressed them in flowchart form.

Comment #2: **Figure 13 to 15, if not individually discussed in detail, are suggested to be moved to appendix.**

**Reply:** Thanks for the opinion. Given the large number of figures in this section, we have placed figures 13 to 15 in the appendix to simplify the manuscript and give a brief description.

Comment #3: **There still remain grammatical errors throughout the manuscript. A thorough proofreading is needed.**

**Reply:** Considering the reviewer's suggestion, we have revised the grammar and words throughout the manuscript to enhance the grammatical accuracy.

Based on the review comments, we rewrote the results analysis and conclusion sections of the manuscript to make the key information of this study clearer and easier to read for the reader. In the case study section, a framework flowchart step-by-step approach was used for the progressive study. The section is more clearly organized overall, from data description and processing to the application of the research methodology. To keep the reader's attention on the tasks conducted with the research objectives, parts of the conclusions and arguments poorly relevant to the research questions were removed from the manuscript. We have revised the wording and grammar of the manuscript and corrected some grammatical errors.

Thanks to the reviewers' professional comments, we could quickly target the problems and make targeted corrections. We tried our best to improve the manuscript and made some changes.

Yours sincerely,

Corresponding author: Lili Yang

E-mail: yangll@sustech.edu.cn

---

## Author Response (AR1)

Dear Editors and Reviewers:

Thank you for your letter and for the reviewers' comments concerning our manuscript entitled "A Multi-strategy-mode-waterlogging-prediction Framework for Urban Flood Depth" (ID: nhess-2022-36). Those comments are all valuable and very helpful for revising and improving our paper, as well as the important guiding significance to our research. We have studied comments carefully and have made the correction which we hope meet with approval. The main corrections in the manuscript and the responses to the reviewer' s comments are as flowing:

Comment RC#1-1: **A flowchart is necessary for the case study as well, as there have been many details regarding data extraction, preprocessing and comparison between methods, etc. in the case study.**

**Reply:** Your comment is constructive, and we believe that a flowchart would more clearly reflect the steps of the case study and allow for a more progressive approach to the results and conclusions. We have modified the case study sections according to the process steps in the framework and expressed them in flowchart form.

**Relevant changes:** In line 329 to 529, we rewrite the implementation process of the case study according to the workflow of the framework, conducting detailed experiments on the case from data integration to truth verification and discussing the results. We changed the structure as follows:

[Figure]

Comment RC#1-2: **Figure 13 to 15, if not individually discussed in detail, are suggested to be moved to appendix.**

**Reply:** Thanks for the opinion. Given the large number of figures in this section, we have placed figures 13 to 15 in the appendix to simplify the manuscript and give a brief description.

**Relevant changes:** In line 775, we have moved Figures 13-15 to the appendix as suggested by the reviewers.

Comment RC#1-3: **There still remain grammatical errors throughout the manuscript. A thorough proofreading is needed.**

**Reply:** Considering the reviewer's suggestion, we have revised the grammar and words throughout the manuscript to enhance the grammatical accuracy.

**Relevant changes:** We have rewritten and proofread the manuscript, corrected grammatical and word errors, and rewritten some sentences. In the *highlight version of the supplemental file* (the line numbers in this response refer only to the highlight version file), the changes made are highlighted.

Comment RC#2-1: **First, the paper should be more clarified and concise. The introduction and literature review sections are lengthy, but fail to identify both the research gap in the current literature and the research questions to be addressed in this paper. I recommend to integrate the two sections, and clarify the research questions based on the literature review.**

**Reply:** We have rewritten the introduction and literature review sections. On the one hand, we focus on the strengths and weaknesses of the previous literature studies, find the urgent research questions and define our research objectives in this way. On the other hand, we mainly focus on flooding risk prediction and exclude the part of the literature on risk identification and risk assessment that is not very relevant. The introduction and literature review sections are more concise and logical.

**Relevant changes:** In line 53 to 67, a more representative literature review of some physical models is added, and some literatures on machine learning that is less relevant are removed.

*"Two of the most well-known hydrodynamic models and the most used models are SWMM (Rossman 2010) and MOUSE (DHI 2016a). Conventional modeling approaches (1D and 1D-1D) can simulate quite accurately the drainage network. However, in cases of major rainfall events, these types of models are not able to simulate inundation depth in built-up areas and to visualize flood extent. Kourtis I. M. presents and assesses two different modeling approaches for the assessment of urban flooding in a small urban catchment located in the center of Athens, Greece (Kourtis et al., 2017). Yu et al. applied a 2D raster-based diffusion-wave model to determine patterns of fluvial flood inundation in urban areas by using high-resolution topographic data and explored the effects of spatial resolution upon estimated inundation extent and flow routing process. The disadvantage of the 2D model is that it is difficult for the raster data model to predict the submerged area changing with time, and the performance of flow process is relatively simplified due to poor description of momentum transfer on a flood plain (D. Yu & Lane, 2006a). But its advantages are also obvious. Compared with the finite element method, finite difference method, and finite volume method, the 2D model is easy to write, with high computational efficiency and simplified calibration (D. Yu & Lane, 2006b). Abedin used SCS-CN method to estimate surface runoff, superimposed Flow direction Grid and Weight Grid to obtain Flow length Grid, and then obtained Travel Time Grid(Abedin & Stephen, 2019). Zhang et al. presented a model using a new three-dimensional (3D) flooding model, which is*

*an unstructured mesh, finite element model that solved the Navier-Stokes equations and developed based on Fluidity (T. Zhang et al., 2015)."*

~~"Bui et al. compared the performances of ANN, SVM, and RF in general applications to floods, whereby RF delivered the best performance (Tien Bui et al., 2015). Ouyang et al. (Ouyang et al., 2016) and Zhang et al. presented a review of the applications of ensemble ML methods used for floods. EPSs were demonstrated to have the capability for improving model accuracy in flood modeling (Zhang et al., 2018). Discrete wavelet transform (DWT) is widely applied in, e.g., rainfall-runoff (Ravansalar et al., 2017), daily streamflow (Guimarães Santos & Silva, 2014), and reservoir inflow. The accuracy of prediction is improved through DWT, which decomposes the original data into bands, leading to an improvement of flood prediction lead times."~~

In the data-driven models paragraph in line 88, the detailed literatures on SVM and SVR were removed:

~~"SVMs are more suitable for nonlinear regression problems, to identify the global optimal solution in flood models (Tehrany, Pradhan, Mansor, et al., 2015). Although the high computation cost of using SVMs and their unrealistic outputs might be demanding. SVM is used to predict a quantity forward in time based on training from past data. Over the past two decades, the SVM was also extended as a regression tool, known as support vector regression (SVR) (Li et al., 2016). Gizaw and Gan (Gizaw & Gan, 2016) developed SVR and ANN models for creating RFFA to estimate regional flood quantiles and to assess climate change impact. The SVR model estimated regional flood more accurately than the ANN model. be a suitable choice for predicting future flood under the uncertainty of climate change scenarios."~~

In line 130, the discussion of the advantages and disadvantages of hybrid machine learning method has been removed.

Comment RC#2-2: **Second, other sections should also be shortened and presented in a more direct way. For instance, I did not get the information I expected from a conclusion section although length is enough. There are 10 tables and 15 figures, exhausting the readers to get the key information.**

**Reply:** Sorry for the misleading content. We have carefully considered review comments, shortened the length of the data processing and result analysis sections, and removed some of the non-core bases for conclusions that may be confusing and misleading to the reader. The conclusions and the basis of the experimental results supporting the conclusions are highlighted. The results of figures 13 to 15 are briefly presented in the text, and the three figures are placed

in the appendix section to make the text more concise and readable.

**Relevant changes:** In order to facilitate readers to accurately access the important information in the article and clearly understand the arguments of the article. We started by first reshaping the case study according to the workflow of the framework, omitting some of the data processing and data analysis processes.

In line 432 to 440, the comparative descriptions of the results in Tables 2 to 6 were rewritten.

[revised manuscript text omitted]

In line 516 to 520, the excellent performance of mode5 is described and the possible reasons are explained in terms of mechanism

*"To sum up, mode (5) performs better than any other modes, indicating that the short-term prediction of waterlogging considering the change of future rainfall trend is more realistic. LR seems to have achieved good prediction results in all the five modes. However, the factor that cannot be ignored is that the original waterlogging depth data is sparse and uneven, which must be resampling interpolation processing. It is necessary to go through actual value test to judge whether LR method is really applicable to prediction."*

In line 595 to 612, the important conclusions of the paper are described in a relatively short space, discussing which configuration becomes the best choice for flooding depth prediction in the short term under the MSMWP framework in terms of accuracy, computational efficiency, and other aspects of the choice of mode, strategy, and algorithm.

*"In this framework, different prediction strategies were discussed and used to predict multiple dimensions of waterlogging. Results show that the mode of expanded-multi-R and multi-D performs better than any other modes; five regression algorithms are more suitable for waterlogging prediction. Recursive and Multi-Output strategies have a better performance and robustness, but MO prediction strategy has not only higher performance but also more efficient."*

Comment RC#2-3: **Third, I cannot understand why the authors did not include elevation data in the methodology, which should be a critical factor in determining urban floods.**

**Reply:** Elevation data is important for flood hazard prediction and simulation because elevation

directly affects surface runoff flow direction and velocity. It helps researchers to delineate catchment areas, determine watersheds and outlets, etc. The above mainly applies to flood risk prediction and simulation based on hydrodynamic methods. The manuscript is mainly devoted to solving the problem of predicting the future waterlogging depth of urban flood-prone points (determined by municipal management based on historical flooding events). It can solve the problem of the temporal distribution of urban flooding. In lines 154-156, we illustrate that the variables affecting the temporal distribution of flooding depth are mainly rainfall and previous moment flooding depth for sensor sites. The elevation data, surface type data, drainage network distribution data, etc. are constant in these flood-prone points. Hence, they can be regarded as static factors in the machine learning black box model, where the input variables are real-time rainfall data and previous waterlogging depth data, and the output variables are future waterlogging depth. Considering that the current urban ponding waterlogging sensors mainly perform limited real-time monitoring functions and lack prediction functions. Combining the historical flooding depth data of these points and implementing the model configuration, training and correction under this framework can enhance the prediction capability of future waterlogging depth at flood-prone points, which is crucial for the government to release early warning information and carry out emergency dispatch in a timely manner.

Comment RC#2-4: **Finally, grammatical errors are throughout the manuscript.**

**Reply:** Considering the reviewer's suggestion, we have revised the grammar and word throughout the manuscript to enhance the grammatical accuracy.

**Relevant changes:** We have rewritten and proofread the manuscript, corrected grammatical and word errors, and rewritten some sentences. In the *highlight version of the supplemental file* (the line numbers in this response refer only to the highlight version file), the changes made are highlighted.

Based on the review comments, we rewrote the results analysis and conclusion sections of the manuscript to make the key messages of this study clearer and easier to read for the reader. In the case study section, a step-by-step workflow approach was also used for the progressive study. The section is more clearly organized overall from data description and processing to the application of the research methodology. Some of the conclusions and their arguments that were not highly relevant to the research questions were removed from the manuscript to allow the reader to better focus on the various tasks that were conducted with the research objectives in mind. We have revised the wording and grammar of the manuscript and corrected some grammatical errors.

Thanks to the editor and reviewers' professional comments, we could quickly target the problems and make targeted corrections. We tried our best to improve the manuscript and made some changes.

Yours sincerely,

Corresponding author: Lili Yang

E-mail: yangll@sustech.edu.cn